# Contribution of traffic-originated nanoparticle emissions to regional and local aerosol levels

Miska Olin[1], David Patoulias[2,3], Heino Kuuluvainen[1], Jarkko V. Niemi[4], Topi Rönkkö[1], Spyros N. Pandis[2,3], Ilona Riipinen[5], and Miikka Dal Maso[1]

[1]Aerosol Physics Laboratory, Tampere University, FI-33014 Tampere, Finland
[2]Department of Chemical Engineering, University of Patras, GR-26504 Patras, Greece
[3]Institute of Chemical Engineering Sciences, Foundation for Research and Technology, GR-26504 Patras, Greece
[4]Helsinki Region Environmental Services Authority (HSY), FI-00066 HSY, Finland
[5]Department of Environmental Science (ACES) and Bolin Centre for Climate Research, Stockholm University, SE-10691 Stockholm, Sweden

**Correspondence:** Miska Olin (miska.olin@tuni.fi)

**Abstract.** Sub-50 nm particles originating from traffic emissions pose risks to human health due to their high lung deposition efficiency and potentially harmful chemical composition. We present a modelling study using an updated EUCAARI number emission inventory, incorporating a more realistic, empirically justified particle size distribution (PSD) for sub-50 nm particles from road traffic as compared with the previous version. We present experimental PSDs and $CO_2$ concentrations, measured in a highly trafficked street canyon in Helsinki, Finland, as an emission factor particle size distribution (EFPSD), which was then used in updating the EUCAARI inventory. We applied the updated inventory in a simulation using the regional chemical transport model PMCAMx-UF over Europe for May 2008. This was done to test the effect of updated emissions in regional and local scales, particularly in comparison with atmospheric new particle formation (NPF). Updating the inventory increased the simulated average total particle number concentrations by only 1 %, although the total particle number emissions were increased to a 3-fold level. The concentrations increased up to 11 % when only 1.3–3 nm-sized particles (nanocluster aerosol, NCA) were considered. These values indicate that the effect of updating overall is insignificant in a regional scale during this photochemically active period. During this period, the fraction of the total particle number originating from atmospheric NPF processes was 91 %; thus, these simulations give a lower limit for the contribution of traffic to the aerosol levels. Nevertheless, the situation is different when examining the effect of the update closer spatially or temporally, or when focusing to the chemical composition or the origin of the particles. For example, the daily average NCA concentrations increased by a factor of several hundreds or thousands in some locations on certain days. Overall, the most significant effects—reaching several orders of magnitude—from updating the inventory are observed when examining specific particle sizes (especially 7–20 nm), particle components, and specific urban areas. While the model still has a tendency to predict more sub-50 nm particles compared to the observations, the most notable underestimations in the concentrations of sub-10 nm particles are now overcome. Additionally, the simulated distributions now agree better with the data observed at locations having high traffic densities. The findings of this study highlight the need to consider emissions, PSDs, and composition of sub-50 nm particles from road traffic in studies

focusing on urban air quality. Updating this emission source brings the simulated aerosol levels particularly in urban locations closer to observations, which highlights its importance for calculations of human exposure to nanoparticles.

## 1 Introduction

Detailed emission inventories are necessary for predictions of air quality and atmospheric composition in general. At present, very few of the standard inventories focus in enough detail on particle number concentrations and size distributions of particles from various sources. Several modelling studies using the regional chemical transport model PMCAMx-UF (Jung et al., 2010) over Europe (Fountoukis et al., 2012; Ahlm et al., 2013; Baranizadeh et al., 2016; Julin et al., 2018; Patoulias et al., 2018) have relied on the pan-European particle number emission (Denier van der Gon et al., 2009; Kulmala et al., 2011) and carbonaceous

aerosol (Kulmala et al., 2011) inventories developed in the EUCAARI (European Aerosol Cloud Climate and Air Quality Interactions) project (the combination of these inventories is referred here as the EUCAARI inventory). The EUCAARI inventory includes emissions from electricity production, industry, road and non-road transport, waste disposal, and agriculture. Paasonen et al. (2016) estimated future projections of particle number concentrations in a global scale using emission inputs based partially on the same inventory, but, e.g., traffic emissions based on the EU FP7 project TRANSPHORM database (Vouitsis

et al., 2013).

While road transport is a significant particle source in areas affected by vehicles, such as in urban environments (Shi et al., 2001; Kumar et al., 2014), the EUCAARI inventory, however, does not fully consider the traffic-originated emissions of the smallest (especially sub-50 nm, in diameter, $D_\mathrm{p}$) particles. This results partially from the fact that only non-volatile particles larger than 23 nm have been selected as the regulated ones in current road transport number emission standards (Giechaskiel

et al., 2012) because measuring them is far more reproducible than of volatile ones. Many of the components of the smallest particles do, however, evaporate when heated. Hence, there are also emissions of particles larger than 23 nm (volatile ones) which are currently unregulated. The emission factors (EFs) of the smallest particles are quite variable across the vehicle fleet due to the nature of the nucleation process—their main origin at least in diesel exhaust—which is very sensitive to several factors, e.g., fuel properties, driving parameters, exhaust after-treatment technology, and environmental parameters

(Keskinen and Rönkkö, 2010). Only emissions of particles larger than 10 nm were estimated in the EUCAARI inventory, because emissions of especially sub-10 nm particles for many emission sources have not been determined with high enough certainty or not determined at all.

Particles formed via a nucleation process are typically observed as a different mode—called nucleation mode—in the particle size distribution (PSD) of the exhaust. Although the nucleation mode particles are formed from primary gaseous emissions after

the exhaust is released from the exhaust pipe, they are modeled similarly to primary emissions in regional or global models because the grid sizes can be kilometers but the nucleation processes occurring in exhaust plumes occur in scales of a few meters at most. In addition to the high level of variation in the concentrations of the smallest particles in vehicle exhaust, PSD measurements with differential mobility particle sizer (DMPS) or scanning mobility particle sizer (SMPS) typically underestimate the concentrations in sub-10 nm size range (Kangasluoma et al., 2020). Furthermore, particles smaller than 3

nm have remained undetected until recent advances of measurement techniques, such as the introduction of the particle size magnifier (PSM), which is capable of detecting particles down to ∼1 nm (Vanhanen et al., 2011). Traffic has recently been shown to be a major source of those previously undetected particles (nanocluster aerosol, NCA) in traffic-influenced areas (Rönkkö et al., 2017).

Sub-50 nm or sub-23 nm particles originating from traffic are not negligible in terms of human health effects: they have higher deposition efficiency in the human respiratory system as compared with larger particles, and can translocate even to the brain (Oberdörster et al., 2004). They also overlap with the sizes of particles formed and grown during atmospheric new particle formation (NPF) events, and have therefore the potential to contribute to the climate effects of aerosols (Kerminen et al., 2018). Such particles form a complex external aerosol mixture influenced by local co-pollution, meteorology, and atmospheric processes (Rönkkö and Timonen, 2019). Anthropogenic emissions overall can also affect greatly on the frequency and intensity of NPF events in urban air (Saha et al., 2018). Additionally, emissions of diesel vehicles can include metal-containing particles, which can be found in a separate size mode from non-volatile partiles near 10 nm (Kuuluvainen et al., 2020). Metallic combustion-originated nanoparticles have also been found to exist in the brains (Maher et al., 2016).

In this study, the EUCAARI inventory has been updated for more realistic, measurement-derived PSDs originating from road transport. PSDs between 1.2 and 800 nm particles measured in a traffic-influenced street canyon in Helsinki, Finland, were incorporated into the inventory in order to better represent real-world particle emissions from vehicles. The updated inventory was then applied in the PMCAMx-UF model, and the effects of updating were studied at different spatial and temporal scales, compared to the observational data, and contrasted with NPF. The simulated period (May 2008) was photochemically relatively active, which elevates NPF to the major source of new particles. This period was chosen because the same period has been simulated in several other related studies as well, providing plenty of comparable data and pre-defined input files for emissions and meteorology. Since the street canyon measurements were performed in 2017—using more recent technologies for PSD measurements—trends of urban aerosol and vehicle emissions were used to scale the determined emissions from 2017 to 2008.

## 2 Experimental data

The original EUCAARI inventory was updated using PSDs and $CO_2$ concentrations measured at the Mäkelänkatu supersite, located in a highly trafficked street canyon in Helsinki, Finland. The street canyon measurements were performed in May 2017 and in May 2018. PMCAMx-UF simulations were done for May 2008 as in the previous PMCAMx-UF studies over Europe (Fountoukis et al., 2012; Ahlm et al., 2013; Baranizadeh et al., 2016; Julin et al., 2018). More recent measurements for determining traffic emissions were used because PSD measurements down to ∼1 nm were unavailable in 2008. Hourly PSD data are also available for several atmospheric measurement stations across Europe for May 2008.

### 2.1 Determining traffic emission factors

The Mäkelänkatu supersite is a continuous measurement site operated by the Helsinki Region Environmental Services Authority (HSY). It is located at a curbside of a highly trafficked (28 000 vehicles per workday) street canyon about 3 km north of the

city center of Helsinki, Finland. About one tenth of the traffic is comprised of heavy-duty vehicles. The detailed information on the supersite and the measurements performed in May 2017 can be found elsewhere (Kuuluvainen et al., 2018; Hietikko et al., 2018; Olin et al., 2020). Additionally, the composition of NCA (volatile and non-volatile fractions), measured at the

Mäkelänkatu supersite in May 2018 (Lintusaari et al.) and the particle compositions (black carbon (BC), sulfate ($SO_4$), and primary organic aerosol (POA) fractions) in diluted exhaust of a diesel bus, obtained from a simulation with an aerosol dynamics model coupled with a computational fluid dynamics (CFD) model (Olin, 2013), were used in splitting the EFs further into chemical compound categories specified by the EUCAARI inventory.

PSDs ($\mathrm{d}N/\mathrm{d}\log D_\mathrm{p}$) were determined with the combination of a particle size magnifier (PSM), two condensation particle

counters (CPCs), and a differential mobility particle sizer (DMPS), as described by Olin et al. (2020). In addition to the study by Olin et al. (2020) taking only a large-particle dilution ratio (DR$= 8.2$) of the used bridge diluter into account, DR is now afterward corrected for very small particles. The correction was done using a DR-vs-$D_\mathrm{p}$ curve determined in an inverse modelling study with CFD (Olin et al., 2019). The corrected DR for the first two size bins (1.2–3 nm and 3–7 nm) are 10.7 and 8.8, instead of the constant value of 8.2.

The concentrations ($N$ in $\mathrm{cm}^{-3}$) of every size bin of the determined PSDs were converted to EFs ($n$ in $1/\mathrm{kg_{fuel}}$) using simultaneous $CO_2$ concentration measurements (examples shown in Fig. S1) in 1 min time resolution, as was done for the NCA concentration by Olin et al. (2020). To express all data in similar time resolution, the PSDs measured with the DMPS in 9 min resolution were interpolated to 1 min resolution before calculating the EFs. Whereas NCA measured at the curbside probably originates from the studied street or via atmospheric NPF, larger particles—having longer atmospheric lifetime—can

be originated also from larger area, including nearby streets or the whole urban area. Nevertheless, due to the fact that linear fitting of the particle concentrations from every size bin against the $CO_2$ concentration is possible (Fig. S1), their relation to the traffic is evident, although all particle sizes may not be originated from the studied street. The calculated EFs are here represented as an emission factor particle size distribution (EFPSD, $\mathrm{d}n/\mathrm{d}\log D_\mathrm{p}$), presented later in Sec. 3.2.2.

## 2.2 Atmospheric measurement stations

Simulation results are compared with the observations from several atmospheric measurement stations across Europe. PSD data from six measurement stations from the EUSAAR (European Supersites for Atmospheric Aerosol Research) network and from the SMEAR (Station for Measuring Ecosystem-Atmosphere Relations) III station in Helsinki were utilized in the model evaluation.

The selected EUSAAR stations (Kulmala et al., 2011) represent different types of locations: Aspvreten, Sweden, and Mace

Head, Ireland, are located in coastal areas, Hyytiälä, Finland, and Vavihill, Sweden, are located in rural continental areas, Ispra, Italy, and Melpitz, Germany, are not in close vicinity of pollution sources but are still affected by traffic emissions. The SMEAR III station in Kumpula in Helsinki, Finland, is located in an urban background area and the nearest busy road (50 000 vehicles per day) is separated from it by a 150 m band of deciduous forest (Järvi et al., 2009). The Kumpula station is less than 1 km away from the Mäkelänkatu station; thus, they are quite comparable. However, the Mäkelänkatu station is much more

affected by traffic because it is located at a curbside of a busy street canyon (Okuljar et al., 2021). They fall inside the same computational grid cell of the PMCAMx-UF model in this regional scale application.

## 3 Simulations

Simulations were performed with the PMCAMx-UF model for 1–29 May 2008, similarly to Julin et al. (2018). The results from the first two days were omitted from the analysis to minimize the effects of uncertain initial conditions. The model was
run with the original and with the updated emission inventory. The effects of traffic emissions and atmospheric NPF were also examined by performing the model runs also without NPF.

### 3.1 Model description

The three-dimensional regional chemical transport model PMCAMx-UF simulates both the size-dependent particle number and chemically resolved mass concentrations (Jung et al., 2010). Vertical and horizontal advection and dispersion, wet and
dry deposition, and gas-phase chemistry descriptions are based on the publicly available CAMx (Comprehensive Air Quality Model with Extensions) air quality model. Aerosol dynamics processes in PMCAMx-UF, NPF, condensation, and coagulation, are modelled using the DMAN (Dynamic Model for Aerosol Nucleation) by Jung et al. (2006). DMAN tracks the aerosol mass and number distributions using the TOMAS (Two-Moment Aerosol Sectional) algorithm (Adams and Seinfeld, 2002), in which particles are logarithmically divided into 41 size bins between 0.8 nm and 10 μm.
This study used the most recent version of the PMCAMx-UF model, used also by Julin et al. (2018). In this version, particles contain 15 chemical components: POA, BC, $SO_4$, ammonium ($NH_4$), five secondary organic aerosol (SOA) components separated according to their volatility, crustal material, nitrate, sodium, chloride, a surrogate amine species, and water ($H_2O$). The model predicts NPF rate from the sum of the rates of three included NPF mechanisms: the cluster kinetic model ACDC (Atmospheric Cluster Dynamics Code, McGrath et al. (2012); Olenius et al. (2013))-based sulfuric acid ($H_2SO_4$)–ammonia–$H_2O$ and
$H_2SO_4$–dimethylamine–$H_2O$ mechanisms and the classical nucleation theory-based $H_2SO_4$–$H_2O$ mechanism (Vehkamäki et al., 2002). The used computational grid covered the European domain with a 36 km × 36 km horizontal grid resolution and 14 vertical layers reaching an altitude of 6 km. More detailed information of the used model version can be found in Julin et al. (2018).

### 3.2 Updating the emission inventory

#### 3.2.1 Extracting the road transport-related particle emissions from the EUCAARI inventory

Hourly gridded particle emissions in the EUCAARI emission inventory are separated into 15 source categories and subcategories. One of the categories is for road transport and it is further separated to four sub-categories: gasoline, diesel, liquefied petroleum gas, and non-exhaust (e.g., from tires or brakes) emissions. Because the particle number emission rates in 41 size bins in a source category-level were not openly available, updating only road transport-related emissions was not straightforward.

The road transport-related emissions were extracted from the inventory—reporting the particle number emissions as a sum of all 15 sources (separated in all size bins and components)—through a Positive Matrix Factorization (PMF) analysis.

The most optimal solution from the PMF analyses was obtained when the inventory was represented with 16 factors, according to the decrease of the normalized error with increasing the number of factors. Due to an inexact nature of PMF, the optimal solution was not obtained with 15 factors even though the inventory has been constructed with 15 sources. Figure S2

presents maps of the monthly mean abundances of all 16 PMF factors. The factors 5, 6, 7, 11, and 12 have features reflecting real traffic patterns. However, Fig. S3 presenting means of diurnal variations of the abundances of the PMF factors in Kumpula/Mäkelänkatu and Melpitz displays that reasonable diurnal cycles for both stations are seen only with the factors 6, 7, and 11. Of these, only the PSD from the factor 6 (Fig. S4) corresponds to the on-road diesel exhaust PSD, presented by Denier van der Gon et al. (2009), which is also a bimodal distribution having the modes at 23 and 57 nm. The road transport-related

source in the original EUCAARI inventory was available as the total particle mass emission rate. Thus, the map and diurnal variation of the particle mass emission rate from the factor 6 were compared with the ones from the EUCAARI inventory (Fig. S5). It can be seen that the map features, diurnal variations, and the level of the values overall are very comparable with some exceptions, such as some ship routes in the factor 6, due to inexactness of PMF. However, marine areas were omitted from the following emission updates.

Finally, the PMF factor 6 was selected to represent the road transport-related sub-category updated in this work. Although the road transport-related emissions in the inventory consist of four sources, only the factor 6, which is presumably only diesel-related, was used in updating the inventory because it was connected to road emission with high certainty. Omitting the other sub-categories (gasoline, liquefied petroleum gas, and non-exhaust emissions) is not significant because the abundances of the other factors are lower compared to the factor 6 and because using this factor already slightly overestimates the mass emissions

(Fig. S5).

### 3.2.2 Emission factor particle size distribution

Figure 1 presents the EFPSD derived from the PSD measurements in Mäkelänkatu. Its shape agrees well with the shape of the difference PSD (background PSD subtracted from the PSD measured when wind blew from the road) from the same experiment reported by Hietikko et al. (2018), with the exception of a slightly higher soot mode in the difference PSD. The agreement

implies that deriving an EFPSD from bin-by-bin calculation of EFs using $CO_2$ concentrations is an acceptable method. The concentration at the first size bin (1.2–3 nm) is calculated as the average (circled dot) of two values: the value (dot) derived from the experiment in 2017 and the value (circle) derived from the experiment in 2018 (Lintusaari et al.). This was due to a reason that the concentration of the first bin was lower than the next bin (3–7 nm) with the year 2017 data. This is unexpected and possibly caused by uncertainties involved in the detection and penetration efficiency corrections for the particles in the first

bin (NCA-sized). The efficiencies of NCA are very low and thus prone to high relative uncertainty. The EF of NCA from the study by Lintusaari et al.—which, in that case, is higher than the EF of the next bin—was utilized because more sophisticated efficiency calculations were performed there and is thus considered more accurate. Particles in the first two size bins simulated

with the PMCAMx-UF model (0.8–1.3 nm) originate only from NPF processes; such particles also cannot be measured using aerosol instrumentation.

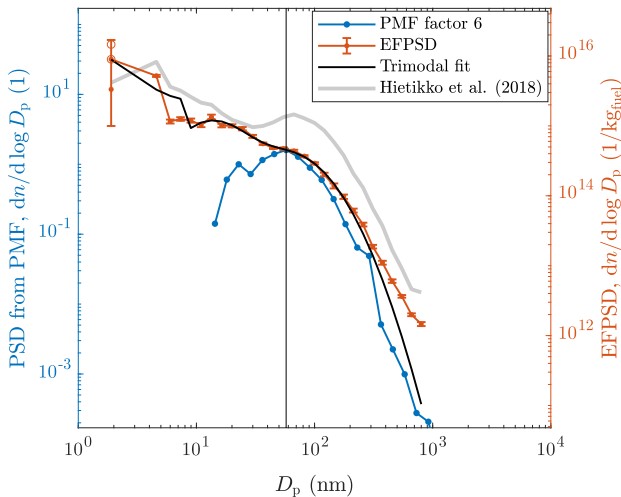

**Figure 1.** PSD from PMF factor 6 and EFs measured at Mäkelänkatu as PSD (EFPSD) together with the trimodal fit on it. The vertical line at 57 nm denotes the highest $D_p$ considered in the updating process and is also the size where the PSDs overlap. The shape of the difference PSD measured at Mäkelänkatu (Hietikko et al., 2018) is also shown for comparison (the data is scaled so that it can be easily compared with the EFPSD data).

The EFPSD, expressed in the unit of $1/\mathrm{kg_{fuel}}$, was converted to correspond to the emission source input of the model, expressed in the unit of $\mathrm{m^{-2}h^{-1}}$ in the following way. The yearly $CO_2$ emissions from road transport in the EU was $7.9 \times 10^{11}\,\mathrm{kg}$ in 2008 (European Environment Agency, 2021). It corresponds to the fuel combustion of $2.5 \times 10^{11}\,\mathrm{kg_{fuel}a^{-1}}$, which was further corrected with the factor of the population count within the simulation domain and in the EU, resulting in the fuel combustion of $5.7 \times 10^7\,\mathrm{kg_{fuel}h^{-1}}$. The EFPSD, determined for the year 2017, expressed as PM2.5 is $0.31\,\mathrm{g/kg_{fuel}}$. However, due to tightened emission regulations, led to introduction of vehicles emitting fewer soot particles (DieselNet, 2021), e.g., by equipping vehicles with a diesel particulate filter (DPF) (Wihersaari et al., 2020), the EF of PM2.5 has been higher in 2008 (EMEP, 2021). Decreasing BC and PM2.5 concentrations in Mäkelänkatu have also been observed from the long-term measurements since 2015 (Barreira et al., 2021; Luoma et al., 2021). The determined EF of PM2.5 was thus estimated to correspond the EF for the year 2008 using the yearly decrease rate of PM2.5, $7.1\,\%\mathrm{a^{-1}}$ (Luoma et al., 2021), resulting in the EF of $0.87\,\mathrm{g/kg_{fuel}}$. That leads to the value of $4.9 \times 10^7\,\mathrm{gh^{-1}}$ for the simulation domain. This value is the same for the hourly emission of PM2.5 obtained from the PMF factor 6, which leads to that the levels of EFPSD and the PSD from PMF match with each other at $D_p$ of 57 nm. The yearly decrease rate of PM2.5 ($7.1\,\%\mathrm{a^{-1}}$) was, however, reported as statistically not a significant trend (Luoma et al., 2021) and also it only covers the trend between years 2015 and 2018. Thus, a trend was also estimated with the data from Kumpula, which fully cover the years between 2008 and 2017. Applying a seasonal

Mann-Kendall test and Sen's slope estimator—as done by Luoma et al. (2021)—to the particle number concentration at 56 nm gives the yearly decrease rate of $4.4\,\%\mathrm{a}^{-1}$ for the years between 2008 and 2017. Since this trend is for Kumpula, the trend for Mäkelänkatu could be around $7.1\,\%\mathrm{a}^{-1}$ because the trends of other quantities for Mäkelänkatu were found to be approximately 2-fold than for Kumpula in the study by Luoma et al. (2021). Additionally, the PM2.5 trend was calculated from the data of yearly (1990–2019) road transport emissions (without road, tyre, and brake wear) in Finland, reported by EMEP (2021). The
decreasing trend calculated for the years between 2008 and 2017 is $6.0\,\%\mathrm{a}^{-1}$, which corresponds relatively well to the trend applied here ($7.1\,\%\mathrm{a}^{-1}$).

Because fuel efficiency has developed during the years, $CO_2$ emissions from road transport have been on different levels in 2008 and in 2017. The method of determining EFs using $CO_2$ concentrations gives the EFPSD with respect to kilograms of fuel combusted. Therefore, it can be applied to any year. However, the total amount of combusted fuel in the computational
grid with respect to time has changed, leading to the need of scaling the time-based particle emission rates—which is the form of the emission input of the model—upwards from year 2017 to year 2008. This scaling has, however, already been performed when the EFs were scaled using the trends of PM2.5, because ambient PM2.5 concentrations have decreased not only due to equipping vehicles with a DPF but also due to the fact that the total amount of fuel combusted has decreased.

The shapes of the PSD from PMF factor 6 and the estimated EFPSD beyond 57 nm agree relatively well, suggesting that the
soot mode was estimated quite accurately already in the original EUCAARI inventory. Because the PSDs of the soot modes lie on similar levels, the emitted particle mass was affected only marginally in the update. The soot mode is assumed to be already estimated well also because exhaust soot measurements have much longer history than measurements of smaller particles. Additionally, the soot particle concentration is not as sensitive to driving and environmental parameters as of smaller particles. 57 nm was selected as the upper limit for which the updating process was applied, i.e., no changes to the original inventory for
$D_\mathrm{p} > 57$ nm was made.

### 3.2.3   Uncertainties involved in updating the emission inventory

Here we elaborate further on the uncertainties involved in representing the road transport-related emissions Europe-wide with a single EFPSD determined from the measurements in Mäkelänkatu in 2017.

Estimating the level of the EFPSD for the year 2008 from the measurements performed in 2017 includes high uncertainty
because the used yearly decrease rate of PM2.5 by Luoma et al. (2021) was determined from the measurements beginning not until 2015 and includes its own uncertainty (including statistically not a significant result). Nevertheless, the scaling of the soot modes was a primary objective here because, hence, the update of the inventory considers only updating the shape of the emitted PSD (below 57 nm), but not its level overall. Additionally, estimating the possible change of the shape of the PSD during the years was not possible. It is, nevertheless, expected that while equipping vehicles with DPFs, soot particle
concentrations are decreased but also the smaller particles may have been decreased. That is because a DPF can filter small particles also—if they are primarily emitted—and because fuel sulfur content has been reduced (DieselNet, 2021), leading to fewer particles formed via sulfur-driven nucleation (Maricq et al., 2002; Kittelson et al., 2008). It should, however, be noted that while the particle emissions from diesel vehicles have been decreased over the last few years, the gasoline vehicle fleet

has begun to emit more particles due to the increased favoring of gasoline direct injection technologies (Awad et al., 2020). On one hand, this increases the uncertainty in estimating the EFPSD for 2008 using the data from 2017; but on the other hand, it provides better estimation on the air quality affected by the modern vehicle fleet.

Vehicle fleets differ among countries, e.g., by fuel selection and by the ages of the vehicles. The average vehicle age in Finland is similar to European average, while diesel vehicles are on average slightly less popular in Finland than in rest of Europe (Eurostat, 2021). It should, however, be noted that averaging of vehicle ages or fuel types over Europe is not the most representative in terms of the average emissions or particle exposure because there are countries having old vehicle fleet with mostly diesel vehicles—a combination with a plenty of soot emissions—but also countries having new vehicle fleet also with mostly diesel vehicles—a combination with the least particle emissions. In addition, there are countries with other possible mixtures of fleet ages and fuel types of vehicles as well.

Particle emissions depend on driving parameters, such as on engine load (Rönkkö et al., 2006). Therefore, particles emitted on an urban street, such as Mäkelänkatu, do not fully represent the particles emitted on other road types, such as on motorways, where higher engine loads are utilized. However, there are signal-controlled intersections on Mäkelänkatu near the measurement site providing also data for emissions with higher engine loads—during accelerations.

Particle emissions depend also on environmental parameters, such as temperature (Mathis et al., 2004; Olin et al., 2019) and radiation (Olin et al., 2020). Therefore, particle emissions can differ between nighttime and daytime. Here, we aim for a first level approximation of PSDs of the emissions using a single EFPSD—for the the most representative average covering the whole vehicle fleet, driving parameters, and environmental parameters in May. Despite this simplification, it is a useful first step in determining the importance of these particles. To our knowledge, in addition to the Mäkelänkatu site, no other location with simultaneous $CO_2$ and PSD measurements down to ∼1 nm is available.

Number-based EFs of especially sub-30 nm particles could be quite different if the emissions were determined from measurements performed at a different location, on a different road-type, and at a different time. In contrast, EFs of particles larger than 30 nm—mainly soot—would possibly differ much less with differing location or time. Nevertheless, the approach in this study still represents the most realistic approximation currently available and it improves the representation of the road traffic-emissions in the original inventory, which excluded all sub-10 nm particles. Emissions of sub-10 nm particles have been applied also in the study by Paasonen et al. (2016), who included a size bin for 3–10 nm particles, based on the TRANSPHORM database (Vouitsis et al., 2013). However, they did not include any modes smaller than 10 nm; thus, this size bin was only an extension from PSDs with larger modes. Kontkanen et al. (2020) compared annual size-binned particle emissions between their estimations from ambient data measured in urban Beijing and the model by Paasonen et al. (2016). They observed that the ambient data suggest significantly more particles in sub-60 nm size range. This is due to the fact that the ambient data represent emissions from a more localized—traffic-influenced—area but also because the smallest particles are omitted from the traffic emissions in the TRANSPHORM database.

### 3.2.4 Parameters of the emission factor particle size distribution utilized in updating the emission inventory

To utilize the determined EFPSD within PMCAMx-UF, it was transformed to the model size bins through a continuous fit (Fig. 1). A trimodal fit consisting of a power law distribution and two log-normal distributions (see the Supplement for the detailed equation) is used because there seems to be features of two log-normal distributions—as typical in vehicle exhaust—but

the smallest particles cannot be fitted very well to any log-normal distribution. A power law distribution fits moderately and is suggested by theory of simultaneous nucleation and growth processes (Olin et al., 2016). The parameters of the fit are presented in Table 1. It is interesting that trimodal size distributions of non-volatile particles—with quite similar particle sizes to the ones found in this study—were also detected in diesel exhaust by Kuuluvainen et al. (2020). They conclude that the mode in the middle is originated from lubricating oil, whereas it is here associated with nucleation-originated particles.

**Table 1.** Mode parameters of the trimodal fit on the measured EFPSD and the estimated particle composition.

| Mode name | Power law | Nucleation | Soot |
|---|---|---|---|
| $n$ ($10^{14}/\mathrm{kg_{fuel}}$) | 115 | 17.2 | 6.44 |
| $D_1$ (nm)[a] | 1.2 | – | – |
| $D_2$ (nm)[b] | 8.0 | – | – |
| $\alpha$[c] | -1.2 | – | – |
| CMD (nm)[d] | – | 13.4 | 59.0 |
| GSD[e] | – | 1.8 | 1.9 |
| BC mass fraction | 0.158 | 0 | 0.688 |
| $SO_4$ mass fraction | 0.128 | 0.152 | 0.064 |
| POA mass fraction | 0.714 | 0.848 | 0.248 |

[a],[b] Diameters of the smallest and largest particles of the mode
[c] Slope parameter
[d] Count median diameter
[e] Geometric standard deviation

The contribution of the road transport-related particle number emissions (from the PMF factor 6, which is presumably related only to diesel vehicles) to the total emissions from all emission sources was averagely 8 % in the original inventory. In updating the inventory, these road transport-related particle number emissions were increased to a 26-fold level, resulting in the increase of the total number emissions to a 3-fold level. Hence, in the updated inventory, the contribution of these road transport-related particle number emissions (from diesel vehicles) to the total emissions becomes 69 %. Due to the lack of all sub-10 nm particle

emissions in the original EUCAARI inventory, sub-10 nm particle emissions in the updated one come exclusively from road transport. By considering only the number concentrations of ultrafine particles (UFP, sub-100 nm particles), the road transport-related emissions were increased to a 28-fold level. This resulted in that the total UFP number emissions were increased by a factor of 3.1.

Vehicle-emitted particles originate primarily via three routes: in-cylinder processes (soot mode, ash particles, non-volatile core), nucleation after the exhaust pipe (nucleation mode), and a less-known source of NCA (power law mode). Therefore, a trimodal fit suits well in separating particle composition between the three sources. However, it should be noted that the vehicle exhaust particle formation is a complex process and this approach is only an approximate. Studies such as Kuuluvainen et al. (2020) and Alanen et al. (2020) divide the non-volatile PSDs of internal combustion engine emissions into three categories, based on PSDs and particle morphology studies, and nucleation mode observed in vehicle exhaust does not always require $H_2SO_4$-driven formation process.

To add particles to the original road transport-related PSD, a selection for their chemical composition was needed. Because measuring chemical composition for sub-50 nm particles is challenging, this study relies on CFD-simulations of particle composition 10 m behind a diesel-fueled bus by Olin (2013). They consist of a situation where a Euro III bus is driving at a speed of 40 km/h with the engine power of $40\%$ of the maximum (see the Supplement for a more detailed description). The CFD-simulations give mass fractions of BC, $SO_4$, POA, and $H_2O$ for the nucleation and soot modes. The road transport emissions in the original EUCAARI inventory consist solely of BC, $SO_4$, POA, and crustal material. Thus, the CFD-simulated mass fractions can be directly utilized in the inventory, with the exception of $H_2O$ which is not included in the emissions due to an equilibrium-type behavior of $H_2O$ dynamics in the model. The chemical composition for the power law mode is determined by, firstly, assuming a fraction of $16\%$ of non-volatile particles (the non-volatile fraction of NCA (Lintusaari et al.)) and, secondly, assuming the nucleation mode composition for the remaining volatile part. The non-volatile part is here lumped together with BC due to the lack of more specific information on its composition and because adding an extra component would have required several modifications to the model code. BC together with the unknown non-volatile part is abbreviated here to BC*. Figure 2 presents the particle chemical composition of the traffic-emitted particles as a function of $D_p$ in the original and in the updated inventory. The composition between 10 and 57 nm is modified to contain more POA and less BC because nucleation mode particles—consisting mainly of POA—were considerably added. Nucleation mode-sized particles were also in relatively low $SO_4$ concentration in the original inventory, but more $SO_4$ is included in the updated inventory. No particles below 10 nm were included in the original inventory. Importantly, the inventory does not include metallic ash particles, that have been reported to contribute particle emissions especially in ultrafine particle size range.

The selection of the CFD-simulations of a diesel-fueled bus for determining chemical composition of particles was further elaborated by examining other related studies as well. Kostenidou et al. (2021) measured chemical composition of particles emitted by different gasoline- and diesel-fueled Euro 5 light-duty vehicles over different transient driving cycles on a dynamometer. Calculated from the reported EFs, the mass fractions of BC, $SO_4$, and POA in the total aerosol were 0.58–0.98, 0.00–0.30, and 0.02–0.15, respectively. Similarly, Pirjola et al. (2019) measured a diesel-fueled Euro 4 light-duty vehicle and reported the BC, $SO_4$, and POA mass fractions of 0.81–0.88, 0.00–0.03, and 0.11–0.18, respectively. These mentioned mass fractions are comparable to the mass fractions in the soot mode from the CFD-simulations (Table 1). However, it should be noted that in the mentioned studies, $SO_4$, and POA were measured using aerosol mass spectrometers, which do not efficiently detect particles smaller than $\sim 50$ nm. Therefore, the composition of the nucleation mode, or especially of the power law mode, is barely covered in the measured compositions and studies related to these compositions are very scarce. According to the

formation principle of nucleation mode particles, they do not contain BC; thus, POA dominates in the mass fractions of the

nucleation mode (Table 1) as it dominates in the mass fractions of the volatile ($SO_4$ and POA) part of the soot mode. Hao et al. (2019) collected PM2.5 particle samples on filters from a highway tunnel in China and reported the BC, $SO_4$, and POA mass fractions of 0.12, 0.09, and 0.34, respectively. These values lie in the range between the mass fractions of the nucleation and soot modes from the CFD-simulations. In conclusion, due to the scarcity of studies on chemical composition of vehicle-emitted particles and because the CFD-simulated mass fractions (of a diesel bus only) are reasonable according to the other studies

(including tailpipe emissions of both gasoline- and diesel-fueled light-duty vehicles and emissions from a real traffic mixture from a road tunnel), the CFD-simulated ones were used here to cover the whole vehicle fleet. In addition, this study primarily focuses on the updating of the shape of the PSD, but not on the exact chemical composition of emitted particles, which was, however, required to be estimated for running the model with the updated inventory.

### 3.3    Simulation results

#### 3.3.1    Comparing simulated particle number concentrations with observations

Particle number concentrations from the PMCAMx-UF simulations were first compared to the ones observed at the measurement stations. Figure 3 presents hourly means of number concentrations of particles smaller than 10 nm ($N_{<10}$) and larger than 10 nm ($N_{>10}$) with the original (orig) and updated (upd) emission inventories. The data of $N_{<10}$ are shown only for the stations that had reliable PSD measurements in sub-10 nm size range. The lower $D_p$-limit in $N_{<10}$ and the upper $D_p$-limit in $N_{>10}$ for

the simulated and the observed values depend on the corresponding limits of the PSD measurements and vary slightly between the stations. There are overestimations in simulated concentrations of particles between 10 and 50 nm and slight underestimations for particles larger than 100 nm ($N_{>100}$) in the previous studies (Baranizadeh et al., 2016; Julin et al., 2018) with the PMCAMx-UF model, possibly due to missing condensable vapors and particle growth mechanisms (Baranizadeh et al., 2016). Even higher overestimations but also underestimations are seen in $N_{<10}$ (Fig. 3a); however, the most notable underestimations

are now overcome when using the updated emission inventory (Fig. 3c). The highest overestimations in $N_{<10}$ still exist, especially for rural locations. In the case of $N_{>10}$, no notable differences can be seen between the original (Fig. 3b) and updated emission inventories (Fig. 3d), except slightly increased—but still underestimated—concentrations in the lowest end of the simulated concentrations.

The agreement and the correlation with the hourly observations and the scatter for $N_{<10}$, $N_{>10}$, and $N_{>100}$ are also pre-

sented in Table 2 in terms of normalized mean bias (NMB), correlation coefficient (R), and normalized mean error (NME), respectively. Whereas the values remain nearly constants for $N_{>100}$ after updating the inventory, NMB values for $N_{>10}$ are further increased. The most significant differences after updating the inventory are observed with the logarithms of $N_{<10}$, for which NMB is increased from $+12\%$ to $+53\%$. Overestimations of the concentrations of the smallest, roughly sub-50 nm, particles—becoming even more substantial after updating the inventory—highlight the possibility of overestimated NPF rates.

On the other hand, overestimation of the simulated $N_{<10}$ can also be perceived as underestimation of the observed $N_{<10}$ due to the inaccuracy (typically underestimating (Kangasluoma et al., 2020)) of PSD measurements in the sub-10 nm size range.

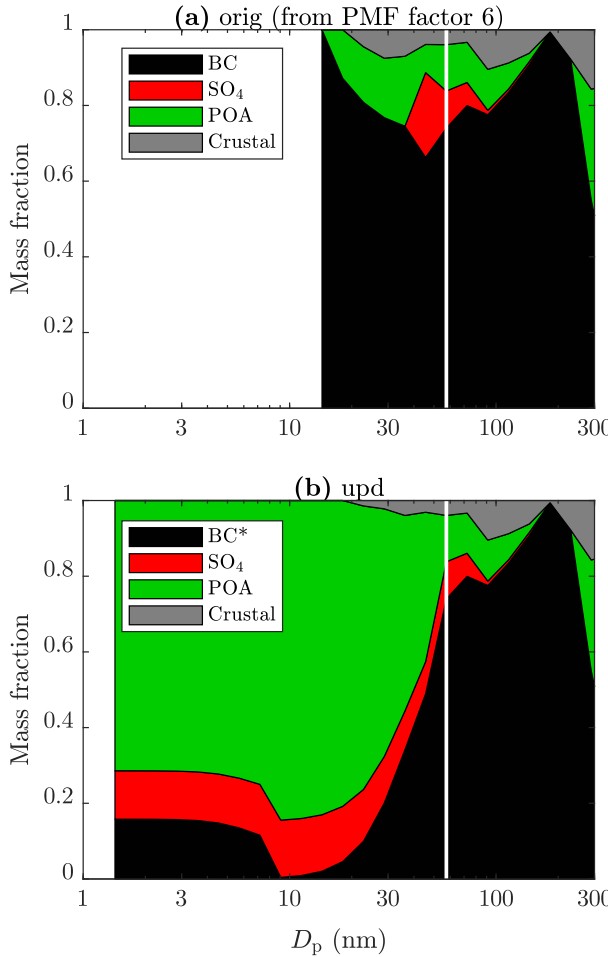

**Figure 2.** Particle chemical composition (**a**) of the PMF factor 6 and (**b**) after updating the emission inventory. The composition of the emitted particles larger than 57 nm (vertical lines) remains unchanged and larger than 300 nm are not shown due to their irrelevance.

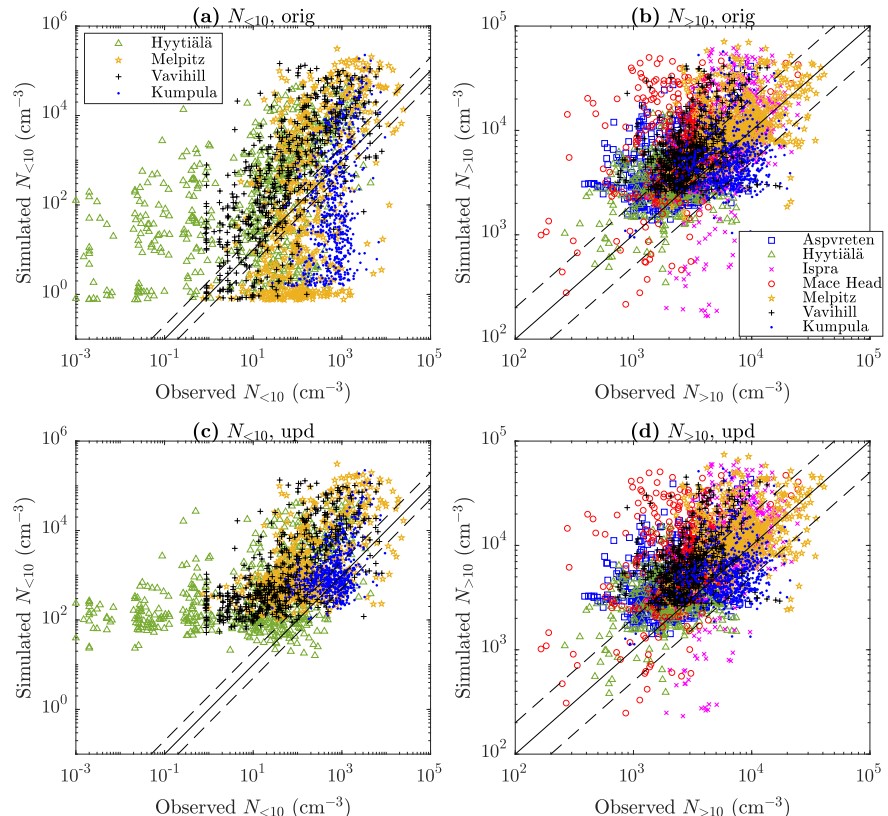

**Figure 3.** Simulated versus observed number concentrations of particles (**a**, **c**) smaller than 10 nm ($N_{<10}$) and (**b**, **d**) larger than 10 nm ($N_{>10}$) at selected measurement stations with (**a**, **b**) the original and (**c**, **d**) updated emission inventory. All data correspond to hourly means for May 2008. The solid diagonal lines represent 1:1 lines and the dashed ones 1:2 and 2:1 lines. See Fig. S6 for a clearer presentation of the data from the stations with the highest traffic influences only (Melpitz and Kumpula).

It should be noted that there are observations (particularly from Hyytiälä and Vavihill) of very low hourly averages of $N_{<10}$ (below $1\,\mathrm{cm}^{-3}$), which may not be of very reliable data due to low counting statistics and which have thus a major role on the disagreement. In contrast to the agreement, improvements for $N_{<10}$ (logarithms) after updating the inventory can be seen in the correlation and in the scatter: R increases from $+0.37$ to $+0.54$ and NME decreases from $64\,\%$ to $58\,\%$, also seen in Fig. 3a,c as overcoming of the most notable underestimations with the updated inventory. In the case of urban locations, even better improvements are seen, e.g., NME decreasing from $42\,\%$ to $16\,\%$ for Kumpula.

### 3.3.2 Effect of updating emission inventory on relative particle concentrations

Figure 4 presents how much the concentrations of 1.3–3 nm ($N_{\mathrm{NCA}}$), 7–20 nm ($N_{7-20}$), and all particles ($N_{\mathrm{tot}}$) change after updating the inventory. The concentrations remain nearly unchanged, especially $N_{\mathrm{tot}}$, but are also stretched out to both di-

**Table 2.** Normalized mean bias (NMB), correlation coefficient (R), and normalized mean error (NME) of the simulated particle number concentrations compared to the observed ones. The values in parentheses denote the values with the original emission inventory. The top values are calculated from the ordinary concentrations and the bottom values from the logarithms of the concentrations. The bold values highlight the most notable differences between the inventories (the best performing in bold).

| | $N_{<10}$ | $N_{>10}$ | $N_{>100}$ |
|---|---|---|---|
| NMB (%) | +1102 (+1066) | +70 (+63) | -12 (-12) |
| R | +0.30 (+0.30) | +0.40 (+0.38) | +0.61 (+0.62) |
| NME (%) | 1142 (1139) | 96 (94) | 49 (49) |
| NMB (%) | +53 (**+12**) | +5.7 (+5.0) | -4.2 (-4.2) |
| R | **+0.54** (+0.37) | +0.50 (+0.47) | +0.65 (+0.66) |
| NME (%) | **58** (64) | 8.6 (8.5) | 10 (10) |

rections, toward decreased and toward increased concentrations. However, all the histograms are slightly displaced from the ratio of one so that increased concentrations are more common. There are also notable extremes in the concentration ratios, especially for NCA (min: 0.0003, max: 4225) denoting that $N_{\mathrm{NCA}}$ was decreased or increased with factors of up to several thousands in certain locations on certain days. Although updating the inventory increases emissions for all particle sizes, it also

leads to decreased concentrations at certain times in certain areas having a high NPF rate. This results via increased primary emissions of particles increasing the condensation and coagulation sinks, which can reduce nucleating gaseous precursors and newly formed particles, respectively, and thus lead to less small particles. Due to a complex relationship between the increases of the sinks and the appearance of small particles, updating the emission inventory can change the particle concentrations in both directions. It is clear that decreased concentrations are related to the connection between NPF and emissions because

simulating with NPF processes switched off results in the situation in which updating the inventory only increases the concentrations.

Figure 5 presents the ratios of the concentration change as maps. In contrast to the histograms in Fig. 4, the ratios in the maps are calculated from the monthly mean values, representing the total aerosol exposure of people living in certain areas. The roughest extremes of the ratios do not exist when examining monthly means but there are still sporadic areas in which

concentrations were decreased or increased by a factor of $\sim 2$ (not shown in the maps). The monthly mean concentrations, especially of $N_{7-20}$, were increased by tens of percents in densely populated areas, especially in Western Europe, but there are also areas having ratios much below or above one over marine areas, such as over the Mediterranean Sea.

The ratios of the concentration change calculated from the monthly means are also presented as mean and median values in Table S1. The values for $N_{<10}$, $N_{<23}$ (totally unregulated vehicle-emitted particles, $D_{\mathrm{p}} < 23\,\mathrm{nm}$), and $N_{<100}$ (UFP,

$D_{\mathrm{p}} < 100\,\mathrm{nm}$) are also shown. Additionally, the values are presented as population density-weighted values using the gridded population count data for 2010 from CIESIN (2018). Updating the emission inventory increased total particle count in Europe

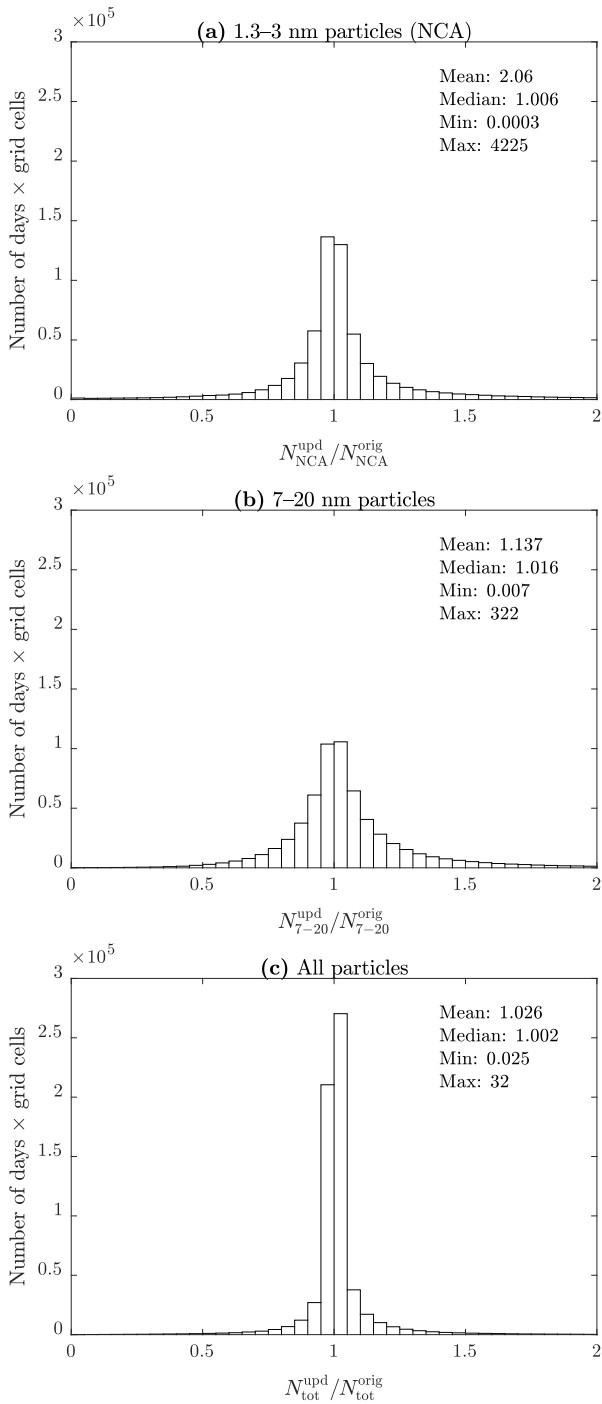

**Figure 4.** Histograms for grid cell-separated ratios of daily means of (**a**) the NCA concentration ($N_{\mathrm{NCA}}$), (**b**) the concentration of 7–20 nm particles ($N_{7-20}$), and (**c**) the total particle concentration ($N_{\mathrm{tot}}$) simulated with the updated and with the original emission inventory. Statistics are also presented numerically on the graphs.

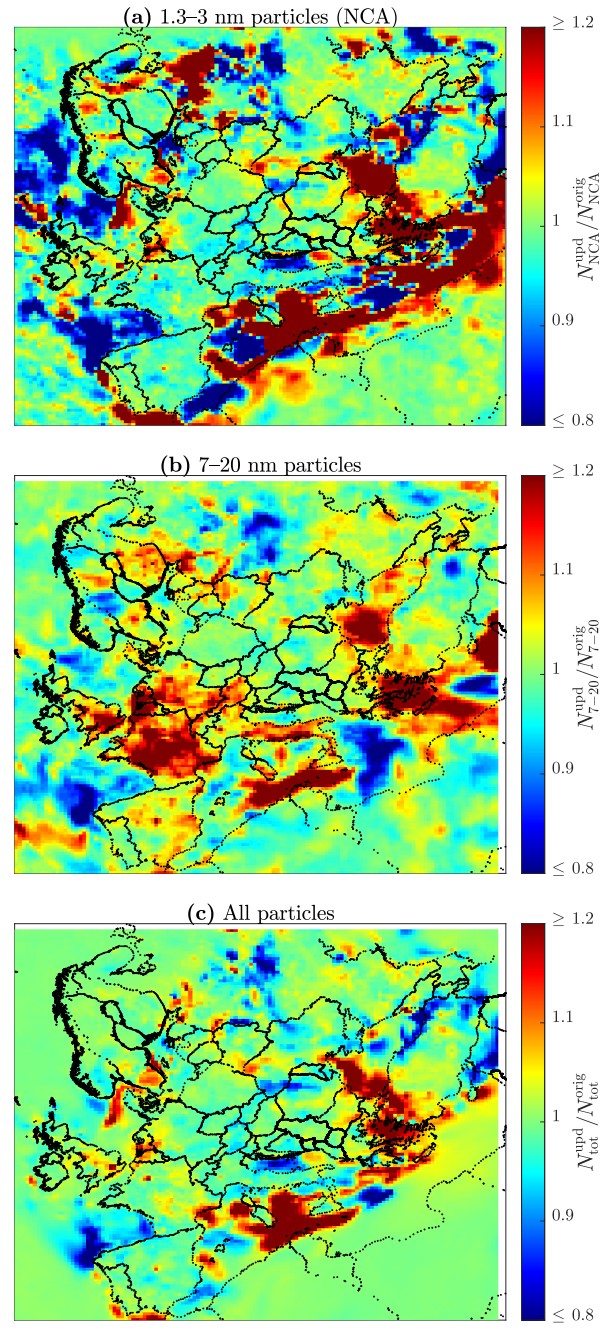

**Figure 5.** Ratios of monthly means of (**a**) the NCA concentration ($N_{\text{NCA}}$), (**b**) the concentration of 7–20 nm particles ($N_{7-20}$), and (**c**) the total particle concentration ($N_{\text{tot}}$) simulated with the updated and with the original emission inventory.

for the whole month with only $1\%$. However, the increase is $2\%$ with using the population density-weighting. That can be interpreted so that the total human exposure on particle number is estimated $2\%$ higher when using the updated inventory compared to the original one. Moreover, the increase is $11\%$ if only NCA-sized particles are considered. The highest differences are observed with considering particles between 7 and 20 nm, for which the population density-weighting gives the mean increase of $10\%$ and the median increase of $4\%$. The latter value can be interpreted so that half of the people within this European domain are, on average, exposed to $N_{7-20}$ with at least $4\%$ more than what would have been estimated using the original inventory.

### 3.3.3 Comparing simulated particle size distributions with observations

The results so far have displayed that the particle concentrations were slightly increased after updating the inventory when the concentrations are averaged over long times and wide areas. The effect of updating the inventory is next examined locally and more temporally, first, by comparing PSDs simulated with the original and with the updated inventory together with the observations. Figure 6 presents monthly means of PSDs at selected measurement stations, separately for mornings (05:00–09:00) and daytime (10:00–14:00). Daytime typically experiences the highest NPF rates, due to the solar radiation cycle, but also high traffic densities. Mornings, instead, have typically even more traffic but not yet solar radiation-ignited NPF. PSDs in the daytime do not differ notably between the original and the updated inventories, with the exception of slightly higher concentrations with the updated inventory in Melpitz and Kumpula for $\sim$5–30 nm particles. Agreement of the daytime PSDs with the observations is fairly good for particles larger than 10 nm, but the overestimation of the simulated particles (or underestimation of the measured particles) smaller than 10 nm can be seen. Melpitz and Kumpula are again different, having higher observed concentrations than the simulated ones. These are locations affected by road traffic, especially Kumpula, and the results hence indicate that traffic emissions may still be underestimated even with the updated inventory. However, it should be noted that the grid cell including the Kumpula station consists of not only urban areas but rural and marine areas too. Therefore, the average concentrations within the grid cell are, indeed, expected to be lower than the concentrations within urban areas only. Additionally, there are a busy airport and harbor areas within a radius of 15 km from the Kumpula station. It is certainly possible that, in addition to road transport, other activities, such as aviation and shipping, can also involve underestimated particle emissions. Hence, other anthropogenic particle emission sources may also need to be addressed better in emission inventories, in order to have the simulated PSDs to agree with the measured ones.

In the case of the morning PSDs, differences between the emission inventories are more notable. The updated inventory predicts levels of sub-30 nm particles up to 3 orders of magnitude higher in areas affected by road traffic (Ispra, Melpitz, and Kumpula) than the original inventory. The use of the original inventory fails to predict PSDs for sub-30 nm particles for the mornings. The updated inventory, instead, gives fairly good agreements for the PSDs when the possible underestimation of PSD measurements for sub-10 nm particles are taken into consideration. People exposed to outdoor air in the mornings in urban areas are exposed to sub-30 nm particles remarkably more than would have been predicted using the original inventory. Furthermore, the differences could be even higher within the urban centers, but the used coarse grid resolution cannot capture the effect in more localized scales.

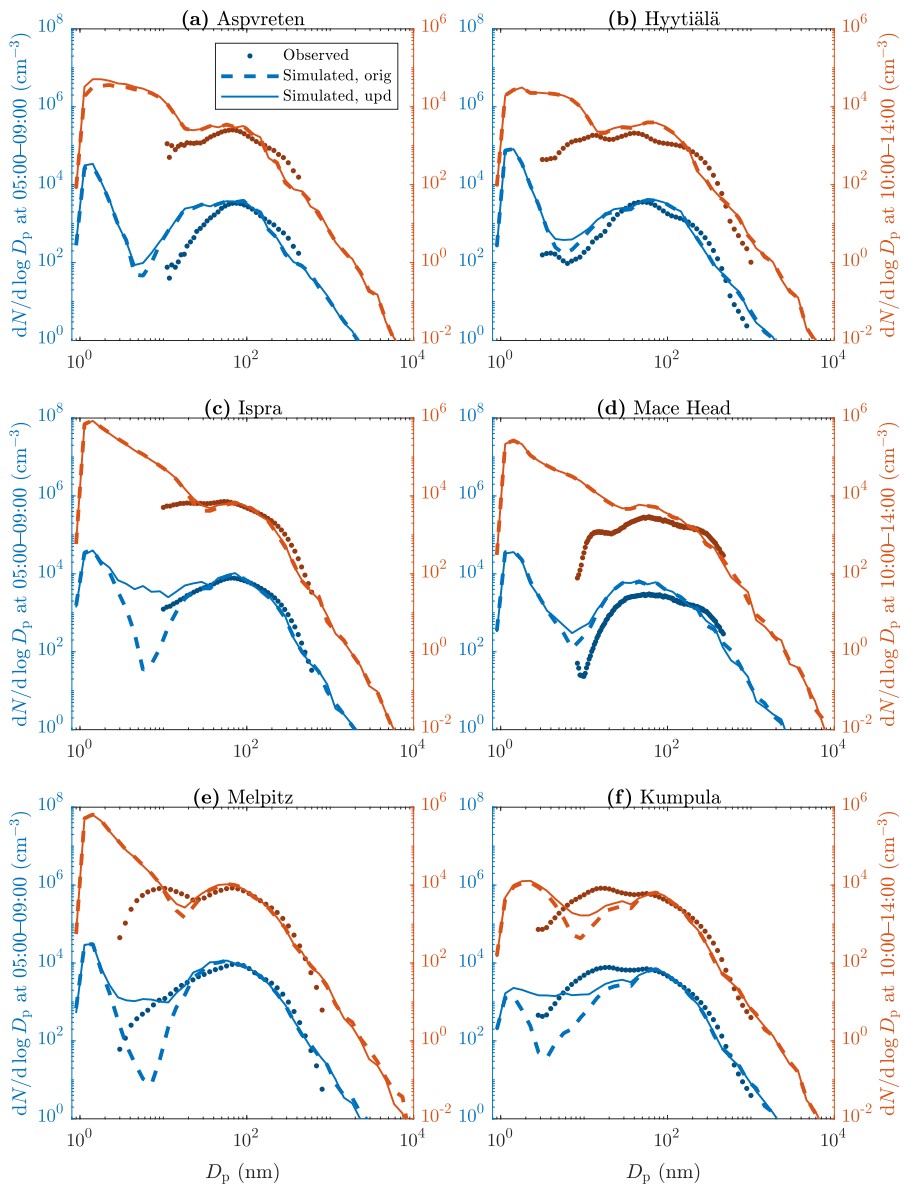

**Figure 6.** Monthly means of PSDs at selected measurement stations (**a–f**) from observations (markers) and from simulations using the original (dashed lines) and the updated (solid lines) emission inventory in the mornings (blue) and in the daytime (red). All times represent local times. Note different axis limits for morning and daytime data.

### 3.3.4  Change of particle composition after updating the emission inventory

Sub-30 nm particles may carry potential health issues because they lie in the range of the highest lung deposition efficiencies ($> 30\%$ for 6–50 nm particles (ICRP, 1994)) and can thus end up in human body, even to the brain via olfactory nerve (Maher et al., 2016). Therefore, they are of high importance, especially in urban areas and if their origin is traffic because emissions from fossil fuel combustion include harmful substances. Simulated particle composition is examined in Fig. 7, as instantaneous composition in Melpitz at 24 May 2008, 09:00–10:00. The selection of this location and time is made to demonstrate how particle composition changes due to updating the inventory while PSD and particle concentration do not significantly change ($N_{\mathrm{tot}}^{\mathrm{upd}}/N_{\mathrm{tot}}^{\mathrm{orig}} = 0.93$, $N_{<10}^{\mathrm{upd}}/N_{<10}^{\mathrm{orig}} = 0.71$). The reason for particle concentrations to even decrease after updating is increased condensation and coagulation sinks, as discussed before. In this case, the total NPF rate was lowered to a level of one third of the rate simulated using the original inventory. However, the sinks are actually $\sim 4\%$ lower with the updated inventory during the time range presented in Fig. 7. Instead, the sinks just before the time range were $\sim 6\%$ higher and even $\sim 10\%$ higher in an adjacent grid cell. The effect of increased sinks with the updated inventory on the appearance of small particles is not always observed within a single time step or grid cell but within later time steps or nearby grid cells instead, due to a history effect and transportation of components between the grid cells.

The composition of sub-30 nm particles was changed so that particularly the mass fractions of POA (and slightly BC/BC*) were increased at the expense of the other components, while the composition of particles larger than 30 nm did not substantially change (Fig. 7a,b). The reason why particularly POA and BC/BC* were increased is because POA and BC* were selected (Sec. 3.2.4) as the main components of the particle emissions of road traffic through CFD-simulations, instead of direct particle composition measurements. Therefore, BC* can also comprise of other non-volatile components, such as of metals, in this context. By examining the change of PSD in Fig. 7c, the effect of updating the inventory seems only minor. Nevertheless, by examining the mass size distributions of certain components in Fig. 7d, it can be seen that POA and BC/BC* masses for sub-10 nm particles were increased significantly from nearly zero-levels even though $N_{<10}$ was decreased. In conclusion, whereas the effect of updating the inventory on PSDs is minor in some locations, masses of potentially harmful components in small—efficiently lung-depositing—particles can still be substantially increased and potentially posing elevated health risks.

### 3.3.5  Comparing the effects of emissions and atmospheric new particle formation on particle size distributions

The effects of primary emissions of particles and atmospheric NPF are examined in Fig. 8, presenting the monthly means of PSDs in Melpitz in the mornings and in Hyytiälä in the daytime. In the mornings in Melpitz, NPF plays a minor role only on PSDs if the updated inventory is used. It is unambiguous due to the location and time range having high traffic densities but not much atmospheric NPF yet. The original inventory, instead, predicts up to 3 orders of magnitude less 2–20 nm particles. In the case of Hyytiälä, the effects are opposite instead. Even the updated inventory does not sufficiently predict the observed aerosol levels (about an order of magnitude lower) when the NPF processes were switched off. Conversely, also the original inventory is sufficient to predict the observed levels and no notable differences are seen between the inventories when the NPF

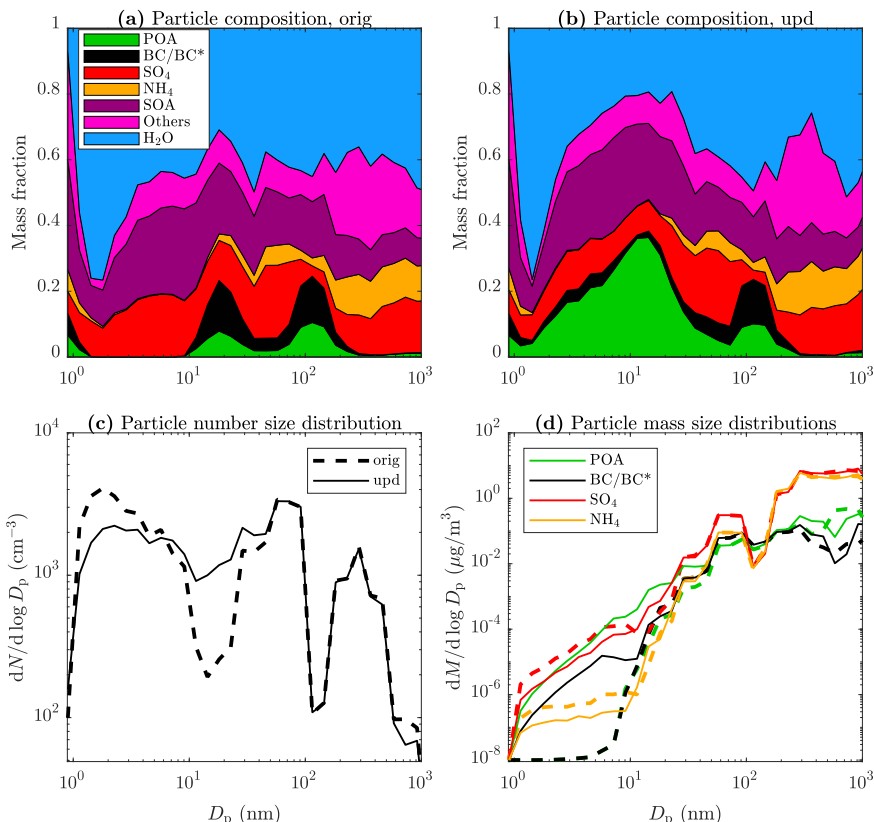

**Figure 7.** Simulated particle composition in Melpitz, at 24 May 2008, 09:00-10:00 local time, using (**a**) the original and (**b**) the updated emission inventory and simulated particle (**c**) number and (**d**) mass size distributions using the original and the updated emission inventory. Others denote the sum of the remaining components, i.e., crustal material, nitrate, sodium, chloride, and the surrogate amine species.

processes were kept on. This was expected, as Hyytiälä is a rural location not greatly affected by road traffic and the daytime is typically associated with atmospheric NPF.

Examining the effects of NPF and emissions within the full European domain displays that the major source of the total particle number is NPF: monthly means of $N_{tot}$ were, in average, decreased with 91 % when the NPF processes were switched off. Without NPF processes, average particle number concentrations increased by 38 % after updating of the inventory although the total particle number emissions increased to a 3-fold level, due to non-linearities in the model, e.g., coagulation. With the NPF processes, the average particle number increase was only 1 %, which is one third of what is expected from the increase of

the emissions if adding particles would not have a lowering effect on NPF rates.

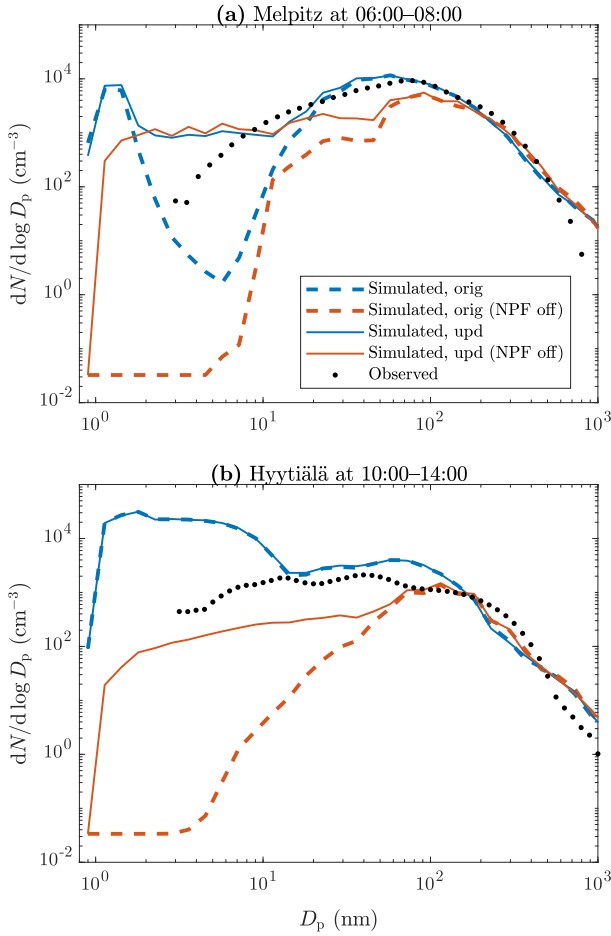

**Figure 8.** Monthly means of PSDs (**a**) in Melpitz at 06:00–08:00 local time and (**b**) in Hyytiälä at 10:00–14:00 local time, according to the simulations using the original and the updated emission inventory, both with the NPF processes kept on and switched off. The observed distributions are also shown.

## 4 Summary and conclusions

Road transport-related particle number emission factors were determined from measurements performed at the curbside of an urban street canyon in Helsinki, Finland. The emission factors were determined separately for every measured particle size bin (1.2–800 nm) and were presented as an emission factor particle size distribution (EFPSD). Deriving an EFPSD from bin-by-bin calculation of emission factors was found an acceptable method based on the agreement with the reported difference between the PSDs measured with wind blowing from the road and from the background direction.

A separate nucleation mode ($CMD = 13\,nm$) and soot mode ($CMD = 59\,nm$) are seen in the derived EFPSD but also a considerable number of particles exists in sub-10 nm size range. Notably fewer sub-50 nm particles and no sub-10 nm particles are included in a road transport-related PSD of the EUCAARI emission inventory, used in several previous studies. This is due to challenges involved in determining emission factors reliably for nucleation mode or smaller particles. In this study, the road transport-related particle emissions of the original EUCAARI inventory were updated using the EFPSD derived here, assuming that it represents the average PSD of the particle emissions from the whole vehicle fleet in Europe.

The PMCAMx-UF model was utilized in simulating aerosol levels for May 2008 over the European domain. The simulations were performed using both the original and the updated emission inventory in order to discover the effect of including the previously partly excluded emissions of sub-50 nm particles. The model overestimates the concentrations of sub-50 nm particles, regardless of the used inventory. Especially sub-10 nm particles are overestimated and the overestimation became even higher when using the updated inventory. The reason for the overestimations may be related to overestimated new particle formation (NPF) or underestimated particle growth but also to possibly underestimated particle concentrations from the PSD measurements, which are known to become inaccurate for particle sizes below $\sim$10 nm. At least, the overestimations of sub-10 nm particles using the updated inventory are not caused by overestimating their emissions because the overestimations were observed also using the original inventory, in which all sub-10 nm particle emissions were excluded. Nevertheless, the greatest underestimations of the model for sub-10 nm particles were overcome and the correlation between the simulated and the observed concentrations was increased, when the updated emission inventory was used.

Ratios of simulated particle concentrations after and before updating the inventory were examined from daily and monthly means of local concentrations. The ratios over and below one were observed while the mean and median values were slightly over one: the predicted concentrations were increased or decreased with a factor of up to several thousands, depending on the examined particle size range, in certain locations and at certain times after updating the inventory. Although particle emissions were only increased in updating the inventory, it resulted also in decreased concentrations due to increased condensation and coagulation sinks leading to less small particles. Examining the ratios from the monthly mean concentrations revealed that, although the total anthropogenic particle number emissions were increased to a 3-fold level, the total particle count in Europe for the whole month was increased by only $1\,\%$ and the total human exposure on particle number with $2\,\%$. The highest mean ratios were observed with considering only 1.3–3 nm particles ($11\,\%$ increase) and the highest human exposures with considering only 7–20 nm particles ($10\,\%$ mean increase and $4\,\%$ median increase). The highest increases were observed in densely populated areas, especially in Western Europe.

The updated inventory predicts up to 3 orders of magnitude higher sub-30 nm particle concentrations during the mornings than the original one in traffic-influenced locations. In those urban locations, simulated PSDs also agree notably better with the observed PSDs.

Because sub-30 nm particles deposit efficiently on the human respiratory system, they pose a significant health risk, especially if their origin is combustion processes emitting harmful substances. Even in cases in which the simulated particle number concentrations did not change markedly, particulate mass of potentially harmful components can increase substantially in the sub-10 nm size range. This results from the substitution of NPF with traffic as the main origin of those particles.

In conclusion, it is important to consider the emissions of sub-50 nm particles from traffic in more detail in chemical transport models, because the previous underestimations (with the original EUCAARI inventory) of particles are located mainly in populated areas and are the greatest for the most efficiently lung-depositing particle sizes. Additionally, the underestimations are especially for particle components having possibly harmful effects on human health. Further investigations on traffic-emitted particles are needed in more local scales than with the coarse grid resolution used in this study. The used model can be operated with a grid resolution of down to, e.g., $1\,\mathrm{km}^2$, provided that an emission inventory for that resolution is available. In addition to road transport, other anthropogenic emission sources, such as aviation and shipping activities, may need to be addressed better in emission inventories, because they may involve underestimated particle emissions as well. Furthermore, estimating long-term particle exposure needs the simulations to be done also for seasons with less photochemical activity, in which the role of traffic emissions may be even more highlighted. The results of this study denote only a lower limit of the contribution of traffic to local aerosol levels due to the coarse grid resolution and due to the selection of the simulation period during which the NPF processes are dominating the particle formation.

*Code availability.* The model code is publicly available at https://github.com/bnmurphy/PMCAMx-UF/

*Author contributions.* MD, IR, MO, SNP, TR, and JVN designed the research. HK performed the measurements. HK and MO analyzed the measurement data. MO and DP updated the emission inventory. MO ran the simulations. MO, IR and MD analyzed the simulation data. MO prepared the paper with contributions from all co-authors.

*Competing interests.* The authors declare that they have no conflict of interest.

*Acknowledgements.* We thank CSC for computational resources. We also thank the authors of the measurement station observation data downloaded from the EBAS and SmartSMEAR databases. Harri Portin and Anu Kousa from the Helsinki Region Environmental Services Authority (HSY) as well as the HSY's AQ measurement team are acknowledged for their valuable work related to the data quality control

and measurements at the Mäkelänkatu supersite. This study has been funded by Finnish Cultural Foundation, by the Academy of Finland through the ACCC Flagship (grant no. 337551), and by Tekes (grant no. 2883/31/2015), HSY, and Pegasor Oy through the Cityzer project.

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
