# Peer review of "Contribution of traffic-originated nanoparticle emissions to regional and local aerosol levels"

_Atmospheric Chemistry and Physics, 2021_

## Referee Comment (RC2)

**Review acp-2021-466**; Miska Olin et al. ; Contribution of traffic-originated nanoparticle emissions to regional and local aerosol levels

The study describes an elegant empirical way to update the only European size-resolved Particle Number inventory and evaluates the results against observations .Ultra fine particles (UFP) and particle numbers (PN) in the atmosphere are not regulated like PM2.5 or PM10. There are no air quality limit values nor obligations to monitor these metrics. As a result our understanding, emission data, measurements, concentrations  and related information is scarce. Clearly more information is needed and the paper is a welcome and original contribution, and fitting for ACP. In my opinion the paper can be published after some corrections and further clarifications have been made.

**Major points**

In the introduction first & second paragraph it would be good to already stress/explain that the EUCAARI inventory by definition considers only particles > 10 nm. The reason being that for many emission sources size-resolved PN measurements are extremely scarce, and EFs for <10 nm are either non-existent or highly variable. Literature is also often not clear about the cut-off. To have a more robust result the EUCAARI inventory was made for size bins of 10 nm and up. However, as pointed pout by the authors, this does not mean that the PN , 10 nm are not important and it is very relevant to investigate this. Therefore "updating" mostly means extending the range of the inventory to include an important but difficult size range.

It is also suggested to explain a bit better that the road transport regulation for PN > 23 nm for non-volatile PN was chosen to have a reproducible measurement. For a standard this is a great advantage. However, a large part of the emitted PN are volatile and it is questionable if this standard for non-volatile PN > 23 nm has any relation or even correlation with the real world total (volatile + non-volatile) size -resolved PN emissions. This relates also to L 301 where some remarks are made about "unregulated vehicle -emitted particles" but volatile particles > 23 nm are also unregulated. So, that is not well-defined in the MS because it suggests road transport particles > 23 nm are regulated but it is only a fraction of these.

Somewhere in the introduction explain more clearly why you choose to do simulations for a year (2008) that is by now 12-13 years in the past. It is not trivial. Many things have changed by now especially in road transport but also e.g. shipping with fuel sulphur regulations.

L186 The updating process was stopped at 57 nm, meaning no changes for Dp > 57 nm. Many studies investigate ultra-fine particles which are defined as < 100 nm. Although not the subject of study it would be good to state in discussion or conclusion what the impact of the update is for anthropogenic UFP emissions, especially for road transport. How much does UFP increase? This may help to put in perspective with other studies.

L190 – why was the scaling done using measurements from 2015 onwards. It would also be possible to use the trend in reported PM2.5 emissions from Finland for road transport exhaust (available at https://www.ceip.at/). Does that give different results? Or do you have a motivation why that would not work?

In the conclusions e.g. around L 415  it would be important to stress / mention that road transport is not the only anthropogenic source where such an update (adding the smaller particles ) would have

an impact. I would expect that for aviation (airports close to a city) and shipping (in the case of port cities) this would further increase the PN emissions. You do not have to add these emission here but it is good to mention that this should be addressed as well. How would further addition influence your model results? Is there room for this or would it lead to overestimating?

L 345 and further: Do you think having substantial sub-10 nm BC particles is realistic? As BC is a product of incomplete combustion it seems not very likely to me? You state it could also be other non-volatile components such as metals but if that is the case, isn't it better to call it non-volatile instead of BC?

Moreover & related, I find this "based on results from one diesel bus" (L82) rather tricky in the light of the whole study. How representative is one Finnish diesel bus for the whole European fleet? Why do the authors feel that is good enough? I think this needs better discussion and motivation.

**Minor points / corrections**

L38 regulated in **road transport** emission standards [it is better to be specific here]

L80 composition of NCA - easier for the reader to have "composition of 1-3 nm sized particles"

L82 chemical composition "obtained from a computational fluid dynamics (CFD) simulation" I find that hard to understand. How can you obtain chemical composition from a CFD simulation? Can you rewrite / explain a bit better?

L143 for  both

L170 and further: The $CO_2$ trick is transparent and elegant but some more discussion or caution on how reliable it is to scale that way to 2008.

L174 " the EF of PM2.5 has probably been higher in 2008."Not probably but certainly – you can check the EEA/EMEP emission inventory guidebook for EFs for different EURO classes.

L 178 is **the** same

L202 – I am not sure how reassuring the "European average" is . This will be mixing e.g. the UK, Sweden, Bulgaria, Portugal etc. The average may not be very representative of what is seen in the different countries if fleet ages and dominant fuel types are highly variable. On average they may cancel out but PN exposure is about the local urban emissions not about the average emission.

Table 1 – Table top row - Please add the size range behind Nucleation and Soot. Easier for the reader.

L229 from the all

Table 2 – caption "(the more intended one being bold)." This is cryptic – please rewrite e.g. you mean the best performing in bold?

L 304 so that  half

L335 the lie on the range = lie in the range

L336 – please rewrite – "travel to human body" is wrong /strange

L 366 after  updating

L384 "previously underestimated emissions of sub-50 nm particles"  I would say "previously partly excluded or partly non-estimated emissions of sub-50 nm particles " The EUCAARI inventory had on purpose a cut-off at 10 nm. In that way it was not "underestimated" but simply not estimated.

L386 "The reason for the overestimations may be related to overestimated new particle formation" Don't you think that must be related is better? Because the model runs with the original inventory do not include any anthropogenic particles < 10 nm but still give the overestimation?.

L391 – cryptic – please rewrite

L396 total **anthropogenic** particle number  (or is that not the case?)

L405 replace fuel-combusting vehicles with combustion processes – I don't think the fact that it is a vehicle is important.

L413 whenever = provided

---

## Author Comment (AC1)

**Final response to the referees' comments for Olin et al.: "Contribution of traffic-originated nanoparticle emissions to regional and local aerosol levels"**

We thank the referees for their insightful comments and have corrected the manuscript according to them.

Referee reports are in *black italic* and authors' responses in blue roman font. **Bold blue** or  fonts highlight changed text parts in some comments. The marked-up manuscript and the Supplement highlighting the changes are included at the end of this file.

**Referee comment 1 (anonymous):**

**1.** *Olin et al. introduce an interesting study on how aerosol particle number emissions described with a previous inventory can be improved by implementing newer emission factor data. The authors derive emission factors for traffic in different size bins from their observations, apply these to improve the description of emissions in an air quality model, estimate the composition of the particles in different size ranges, compare the original and updated modelled particle size distributions and compositions with observations, and discuss the improvements of the update in terms of human health impacts. The topic is timely and the results are interesting and would deserve to be published in ACP, but in the current version the description of the methods and stepwise presentation of results is not adequate for fully understanding the results.*

**2.** *I list first my major comments related to the methods and their illustration, and below this, more detailed comments related to the text. If the authors can satisfactorily reply to these comments and modify the manuscript accordingly, I can recommend the publication in ACP.*

*Major comments:*

**3.** *The authors do not present any figures on the determination of EFPSD, which is one corner stone of this study. They refer to their earlier study in which a similar method was applied for sub-3 nm particles. However, I would assume that the determination of emission factors for larger particles is not as simple, due to their longer lifetimes which causes more varying back-ground concentrations. Where one can expect that sub-3 nm particles at the kerbside are fresh particles either from the traffic or from NPF, larger particles may originate from sources further away and their concentration can be expected to be less sensitive to nearby sources, especially with time resolution as low as 9 minutes. The authors also conclude that the derived EFPSD agrees with the one reported by Hietikko et al. (2018) and that this implies the method used in this article is acceptable. To evaluate the acceptability, the reviewers and the readers need to see how the data look like.*

Figure FR1 presents examples of determining EFs for three different particle size bins (geometric mean diameters of 1.9, 16, and 55 nm). The lowest one corresponds to sub-3 nm particles. It can be seen that linear behavior of the particle concentrations against the $CO_2$ concentrations is observed both for the sub-3 nm particles and for the larger particles too. Longer lifetimes of larger particles can cause that they are originated not from the studied street but from a larger area, such as from the nearby streets or from the whole urban area. Nevertheless, the linear behavior proves that there are particles which can be connected to traffic emissions, although they might not be originated from the studied street. Fig. FR1 is now added to the Supplement (Fig. S1) and the following text to the manuscript:
   "Whereas NCA measured at the curbside probably originates from the studied street or via atmospheric NPF, larger particles—having longer atmospheric lifetime—can be originated also from larger area, including nearby streets or the whole urban area. Nevertheless, due to the fact that linear fitting of the particle concentrations from every size bin against the $CO_2$ concentration is possible (Fig. S1), their relation to the traffic is evident, although all particle sizes may not be originated from the studied street."

[Figure]

**Figure FR1.** Examples of determining emission factors bin-by-bin for three different particle size bins, similarly to the method by Olin et al. (2020). Size-binned particle number concentration data ($dN/d\log D_p$) are averaged within $CO_2$ concentration ($[CO_2]$) bins (circle diameters represent the amount of data used in the averaging). Linear fitting (using the circle diameters as weighting factor) is performed over the averaged data (separately for all 28 measured size bins). The slopes of the linear fits converted to kilograms of fuel combusted are marked in the figure.

[Figure]

**Figure FR2.** The shape of the difference PSD measured at Mäkelänkatu (Hietikko et al., 2018) added for comparison (the data is scaled so that it can be easily compared with the EFPSD data).

For better evaluation of the acceptability of deriving an EFPSD from the measured PSDs, the difference PSD (background PSD subtracted from the PSD measured when wind blew from the road) from Hietikko et al. (2018) is now added to Fig. 1 (shown here in Fig. FR2).

**4.** *The authors use PMF on the diurnal patterns of the EUCAARI emission inventory to extract the contribution of traffic. Does the detailed category-level specification of emissions not exist anymore? Have the producers of the emission data (Denier van der Gon and others at TNO) been contacted to inquire for such specification? If they have been contacted and the specification does not exist, this should be clearly stated and the personal communication to TNO could be used as a reference. Otherwise, I would strongly recommend the authors to reconsider making this contact. The study would seem much more exact or accurate if the original traffic emission output data could be applied.*

We agree that the original traffic emission data would be more exact and accurate. However, we were forced to extract the spatial and temporal patterns and the binned particle emissions and their compositions using the PMF approach because they are not available separately for different source categories, neither at least openly nor at least without significant reprocessing steps by experts not involved in this project. TNO has been contacted. The sentence "Due to the unavailability of the emission rates in a source category-level, updating only road transport-related emissions was not straightforward." is now replaced with the sentence "Because the particle number emission rates **in 41 size bins** in a source category-level were not **openly** available, updating only road transport-related emissions was not straightforward." to highlight the need for the binned emissions and the availability of the original emissions.

Nevertheless, we were able to compare the PMF factor 6 (considered here the road traffic-related source) to the road traffic-related source of the original inventory as the total particle mass emission rates because they were available in a source category-level for the inventory too. Figure FR3 presents the comparison as maps and diurnal variations. It can be seen that the spatial (maps) and temporal (diurnal variations) patterns agree well, with some exceptions, such as some ship routes and slightly higher rates with the PMF factor 6. Therefore, the PMF approach does not cause significant errors to the simulation results.

This discussion is now included in the manuscript (Sec. 3.2.1), Fig. FR3 is now added to the Supplement, and the map and diurnal variation figures (Fig. S2 and S3) in the Supplement are now updated to mass-based variables. Additionally, the map and diurnal variation plot are now removed from Fig. 1 because they are now compared with Fig. FR3 in the Supplement.

**5.** *The authors use CFD modelling for determining the composition of European wide aerosol emissions from traffic. If I understood correct, this CFD model result is based on one diesel bus. Since the CFD model is described only in (non-peer reviewed) MSc thesis (in Finnish), the description of the model and main results should be given in the article. Part of this could be included in the supplementary material. The composition results are not compared to any previous article and the composition is not discussed in the introduction. In the current form, the part on composition should not be published in ACP.*

CFD-modelling was used for determining the chemical composition of particles emitted by road traffic due to a scarcity of related studies, especially when considering particles smaller than 50 nm, which are inefficiently detected with, e.g., an aerosol mass spectrometer. In this first level approximation for updating the road traffic-emitted PSD using a single PSD for the whole European vehicle fleet as in the case of the original inventory as well, the composition was also approximated using a single composition for the whole fleet although it is based on a diesel-fueled bus only. This assumption is now further discussed (and found reasonable) in the manuscript using results from other related studies with the following paragraph added to Sec. 3.2.4:

"The selection of the CFD-simulations of a diesel-fueled bus for determining chemical composition of particles was further elaborated by examining other related studies as well. Kostenidou et al. (2021) measured chemical composition of particles emitted by different gasoline- and diesel-fueled Euro 5 light-duty vehicles over different transient driving cycles on a dynamometer. Calculated from the reported EFs, the mass fractions of BC, $SO_4$, and POA in the total aerosol were 0.58–0.98, 0.00–0.30, and 0.02–0.15, respectively. Similarly, Pirjola et al. (2019) measured a diesel-fueled Euro 4 light-duty vehicle and reported the BC, $SO_4$, and POA mass fractions of 0.81–0.88, 0.00–0.03, and 0.11–0.18, respectively. These mentioned mass fractions are comparable to the mass fractions in the soot mode from the CFD-simulations (Table 1). However, it should be noted that in the mentioned studies, $SO_4$, and POA were measured using aerosol mass spectrometers, which do not efficiently

[Figure]

**Figure FR3.** Monthly means of the particle mass emission rates (**a**,**c**) from the road transport-related source in the original EUCAARI inventory and (**b**,**d**) from PMF factor 6 (**a**,**b**) as maps and (**c**,**d**) as diurnal variations in Kumpula/Mäkelänkatu, Finland, and Melpitz, Germany.

detect particles smaller than $\sim 50$ nm. Therefore, the composition of the nucleation mode, or especially of the power law mode, is barely covered in the measured compositions and studies related to these compositions are very scarce. According to the formation principle of nucleation mode particles, they do not contain BC; thus, POA dominates in the mass fractions of the nucleation mode (Table 1) as it dominates in the mass fractions of the volatile ($SO_4$ and POA) part of the soot mode. Hao et al. (2019) collected PM2.5 particle samples on filters from a highway tunnel in China and reported the BC, $SO_4$, and POA mass fractions of 0.12, 0.09, and 0.34, respectively. These values lie in the range between the mass fractions of the nucleation and soot modes from the CFD-simulations. In conclusion, due to the scarcity of studies on chemical composition of vehicle-emitted particles and because the CFD-simulated mass fractions (of a diesel bus only) are reasonable according to the other studies (including tailpipe emissions of both gasoline- and diesel-fueled light-duty vehicles and emissions from a real traffic mixture from a road tunnel), the CFD-simulated ones were used here to cover the whole vehicle fleet. In addition, this study primarily focuses on the updating of the shape of the PSD, but not on the exact chemical composition of emitted particles, which was, however, required to be estimated for running the model with the updated inventory."

The sentence:

"Because measuring chemical composition for sub-50 nm particles is challenging, this study relies on CFD-simulations of particle composition 10 m behind a diesel-fueled bus by Olin (2013)."

in the preceding paragraph is also now replaced with the sentences:

"To add particles to the original road transport-related PSD, a selection for their chemical composition was needed. Because measuring chemical composition for sub-50 nm particles is challenging, this study relies on CFD-simulations of particle composition 10 m behind a diesel-fueled bus by Olin (2013). They consist of a situation where a Euro III bus is driving at a speed of 40 km/h with the engine power of 40 % of the maximum (see the Supplement for a more detailed description)."

and the Supplement now includes a section "Description of the CFD-simulations used to determine chemical composition of the emitted particles", in which the basics and the applied results of the CFD-simulations are described.

Composition is not discussed in the introduction because it is not of the main interest of this study.

*Minor comments:*

**6.** *Lines 33-34: Traffic emissions in Paasonen et al (2016) are not based on EUCAARI inventory, but on EU FP7 project TRANSPHORM.*

That's true. The text is now updated to read: "Paasonen et al. (2016) estimated future projections of particle number concentrations in a global scale using emission inputs based **partially** on the same inventory**, but, e.g., traffic emissions based on the EU FP7 project TRANSPHORM database (Vouitsis et al., 2013**)."

**7.** *Lines 80-85: is the composition of diesel bus exhaust assumed for the whole fleet? Some words about how this assumption may bias the results.*

Please refer to the answer of the comment 5.

**8.** *Line 92: Some words about how the interpolation may bias the results. Why not using simply 9 min averages of CO2 as well?*

Interpolation of the DMPS data to 1 min time resolution was used because other data was in 1 min resolution too. However, we tested the other method too: keeping the DMPS data in 9 min resolution and averaging the $CO_2$ data to 9 min resolution. This did not have significant differences on the obtained EFPSD: values were slightly ($\sim 6\,\%$) higher but the uncertainties were at least 2-fold compared to the interpolation method. Therefore, we selected the interpolation method.
  The sentence:
  "The PSDs measured with the DMPS in 9 min time resolution were interpolated to 1 min resolution before calculating the EFs."
  is now updated to read:
  "To express all data in similar time resolution, the PSDs measured with the DMPS in 9 min resolution were interpolated to 1 min resolution before calculating the EFs."

**9.** *Lines 100-103: some references to the observation sites should be included.*

The reference to the article by Kulmala et al. (2011) is now added as the reference for the EUSAAR stations.

**10.** *Line 106: Could refer to recent Okuljar et al. article.*

The reference to the article by Okuljar et al. (2021) is now added to this sentence.

**11.** *Lines 145-155: Why do you not investigate further factors 7 and 11, which both seem to be related to rush hours? How is their size distribution and is there a good reason to exclude them? At least factor 7 would contribute significantly to overall result.*

It is true that also the factor 11 seems to be related to rush hours, but it was omitted due to its PSD (Fig. FR4) and map features. Maps and diurnal variation plots are now expressed in mass-based variables, instead of number-based variables, because they are now compared to the ones from the road traffic-related source of the original inventory (Fig. FR3), which is available as the total particle mass emission rate.
  The factor 7 was also omitted due to its PSD and because the factor 6 seems to be enough to represent the whole road transport-source category (seen as slightly higher values in Fig. FR3d).
  This discussion is now included in the manuscript (Sec. 3.2.1), Figs. FR4 and FR3 are now added to the Supplement, and the map and diurnal variation figures (Fig. S2 and S3) in the Supplement are now updated to mass-based variables.

[Figure]

**Figure FR4.** Particle size distributions obtained from PMF factors 6, 7, and 11.

**12.** *Line 148: Since the Denier van der Gon -report is not available without request from the project office, the count mean diameter and/or other features of their PSD should be listed.*

It is now mentioned in the manuscript that "the on-road diesel exhaust PSD, presented by Denier van der Gon et al. (2009)" is "a bimodal distribution having the modes at 23 and 57 nm".

**13.** *Line 176: Luoma et al seem to have the trend calculated mainly for periods 2015-2019. They also mention that the applied trend, -7.1 %/a for PM2.5, is the only trend they could not determine a statistically significant trend. Can you somehow justify extending similar trend to 2008?*

Although statistically not a significant trend, the trend for PM2.5 was applied because the primary objective was to scale the soot modes onto the same levels. In this way, we were able to consider only updating of the shape of the emitted PSD and to omit the updating of the level of emissions overall. This is now clarified in the manuscript; additionally, an estimation of the realism of the trend of $-7.1\,\%\mathrm{a}^{-1}$, using the trend of the concentration of 56 nm-sized particles measured in Kumpula between years 2008 and 2017 and the trend of road transport-emitted PM2.5 in Finland reported by EMEP (2021) for years between 2008 and 2017, is now added.

    The following text is now added to Sec. 3.2.2:

    "The yearly decrease rate of PM2.5 ($7.1\,\%\mathrm{a}^{-1}$) was, however, reported as statistically not a significant trend (Luoma et al., 2021) and also it only covers the trend between years 2015 and 2018. Thus, a trend was also estimated with the data from Kumpula, which fully cover the years between 2008 and 2017. Applying a seasonal Mann-Kendall test and Sen's slope estimator—as done by Luoma et al. (2021)—to the particle number concentration at 56 nm, measured by a DMPS in Kumpula, gives the yearly decrease rate of $4.4\,\%\mathrm{a}^{-1}$ for the years between 2008 and 2017. Since this trend is for Kumpula, the trend for Mäkelänkatu could be around $7.1\,\%\mathrm{a}^{-1}$ because the trends of other quantities for Mäkelänkatu were found to be approximately 2-fold than for Kumpula in the study by Luoma et al. (2021). Additionally, the PM2.5 trend was calculated from the data of yearly (1990–2019) road transport emissions (without road, tyre, and brake wear) in Finland, reported by EMEP (2021). The decreasing trend calculated for the years between 2008 and 2017 is $6.0\,\%\mathrm{a}^{-1}$, which corresponds relatively well to the trend applied here ($7.1\,\%\mathrm{a}^{-1}$)."

    and the following text to Sec. 3.2.3:

    "Nevertheless, the scaling of the soot modes was a primary objective here because, hence, the update of the inventory considers only updating the shape of the emitted PSD (below 57 nm), but not its level overall."

**14.** *Lines 216-218: While this is possibly the case, this sentence should be reconsidered when the Paasonen et al. (2016) paper is notified to be updated in terms of traffic emissions from EUCAARI to TRANSPHORM. One way to discuss the representativeness of Paasonen et al. emissions may be to reflect their comparison to emission size distribution calculated from long-term observations in Kontkanen et al. (2020, https://doi.org/10.5194/acp-20-11329-2020).*

The suggested discussion is now added to the manuscript with the following text after the sentence in question:
   "Emissions of sub-10 nm particles have been applied also in the study by Paasonen et al. (2016), who included a size bin for 3–10 nm particles, based on the TRANSPHORM database (Vouitsis et al., 2013). However, they did not include any modes smaller than 10 nm; thus, this size bin was only an extension from PSDs with larger modes. Kontkanen et al. (2020) compared annual size-binned particle emissions between their estimations from ambient data measured in urban Beijing and the model by Paasonen et al. (2016). They observed that the ambient data suggest significantly more particles in sub-60 nm size range. This is due to the fact that the ambient data represent emissions from a more localized—traffic-influenced—area but also because the smallest particles are omitted from the traffic emissions in the TRANSPHORM database."

**15.** *Lines 220-227: An equation including the three different modes, or at least better explanation, should be given. I am not entirely sure if I understand what the trimodal fit here means. The authors often refer to previous studies in a way that even basic understanding of what is done in this study is not possible to get without reading the other articles (same holds for the determination of emission factor, see my first major comment).*

We agree that some useful detailed information were missing in the manuscript. This issue of understanding of the trimodal fitting is now clarified by adding a note "(see the Supplement for the detailed equation)" to the text and by adding the detailed equation to the Supplement.

**16.** *Lines 228-232: Would be good to mention here also that PMF 6 is presumably related to diesel particles only. And that NCA emissions are not described for other sources in EUCAARI inventory.*

That's a good idea. Not only NCA emissions are described only for road transport in the updated inventory, but actually all sub-10 nm particles. The paragraph is now updated to:
   "The contribution of the road transport-related particle number emissions (from the PMF factor 6**, which is presumably related only to diesel vehicles**) to the total emissions from  all emission sources was averagely 8 % in the original inventory. In updating the inventory, the**se** road transport-related particle number emissions were increased to a 26-fold level, resulting in the increase of the total number emissions to a 3-fold level. Hence, in the updated inventory, the contribution of the**se** road transport-related particle number emissions **(from diesel vehicles)** to the total emissions becomes 69 %. **Due to the lack of all sub-10 nm particle emissions in the original EUCAARI inventory, sub-10 nm particle emissions in the updated one come exclusively from road transport.**"

**17.** *Line 275: Some references required for the underestimation of PSD measurements in sub-10 nm size range.*

A reference to the article by Kangasluoma et al. (2020) is now added in this sentence. Additionally, the reference to the article by Olin et al. (2019) in a sentence also on the underestimation of PSD measurements on line 48 is now replaced with the reference to the article by Kangasluoma et al. (2020), representing a wider study on device comparison in sub-10 nm size range.

**18.** *Line 281-282. It would be interesting to see separate figures for the different sites, especially to those where one would expect the traffic emissions to play a big role in <10 nm concentrations (Kumpula and possibly Melpitz).*

The following sentence is now added to the caption of Fig. 3:

"See Fig. S6 for a clearer presentation of the data from the stations with the highest traffic influences only (Melpitz and Kumpula)."

and Fig. FR5 is now added to the Supplement. From this new figure, the stations with the highest traffic influences only (Melpitz and Kumpula) are more clearly distinguished.

[Figure]

**Figure FR5.** Simulated versus observed number concentrations of particles (**a, c**) smaller than 10 nm ($N_{<10}$) and (**b, d**) larger than 10 nm ($N_{>10}$) at the stations with the highest traffic influences (Melpitz and Kumpula) with (**a, b**) the original and (**c, d**) updated emission inventory. All data correspond to hourly means for May 2008. The solid diagonal lines represent 1:1 lines and the dashed ones 1:2 and 2:1 lines.

**19.** *Lines 290-293: Increasing condensation sink is a plausible reason for decreasing modelled concentrations, but do you have any evidence of that being the reason? N>100 does not seem to change (Table 2), but how much does N>50 change? Or could you draw a map of change in coagulation sink as well? Also later, in Fig. 7, it is difficult to understand the difference in sink: Fig 7c shows very similar size distribution for original and updated model run in sizes >30nm, actually with higher concentration in original run in 40-50 nm and 500-600 nm size ranges. It looks like the sink in the original run in 7c is higher than in the updated run, whereas the interpretation of the differences in lines 341-343 suggests the opposite.*

It is very likely that increased condensation and coagulation sinks affect the concentrations via decreased NPF rates so that the concentrations also decrease after updating the inventory. There is an evidence of at least NPF being connected to the decreasing concentrations: by simulating with the NPF processes switched off, updating the inventory only increases the concentrations. This discussion (including coagulation sink mentioned) is now included in the manuscript by replacing the text:

"This results via increased primary emissions of particles increasing the condensation sink, which can reduce nucleating gaseous precursors and thus lead to lowered NPF rates. Due to a complex relationship between the increase of the condensation sink and the decrease of the NPF rate, updating the emission inventory can change the particle concentrations in both directions."

with the text:

"This results via increased primary emissions of particles increasing the condensation **and coagulation** sink**s**, which can reduce nucleating gaseous precursors **and newly formed particles, respectively,** and thus lead to **less small particles**. Due to a complex relationship between the increas**es** of the sink**s** and the **appearance of small particles**, updating the emission inventory can change the particle concentrations in both directions. **It is clear that decreased concentrations are related to the connection between NPF and emissions because simulating with NPF processes switched off results in the situation in which updating the inventory only increases the concentrations.**"

and by mentioning coagulation sink too in the discussion of Fig. 7 and in the conclusions.

As the change in the values for $N_{>100}$ with updating the inventory in Table 2, the change in $N_{>50}$ is also minor. It should be noted that those values correspond to the data from the selected measurement stations for the whole month, but the decreasing concentration effect is better observed when examining in shorter timescales. Figure FR6 illustrates the maps for the ratios of monthly means of the condensation and coagulation sinks. By comparing these maps to the maps in Fig. 5, it can be observed that the sinks become higher mainly over the ground, in average, but there are several areas with decreased concentrations over the seas. Because higher concentrations basically lead to higher sinks, the increased sinks and the decreased concentrations are not seen at the same areas in averaged maps. This highlights that the effect of adding more particle emissions can lead to decreased concentrations too is a phenomenon which occurs within large areas.

The case in Fig. 7 does actually have $\sim 4\%$ lower condensation sink with the updated inventory as you interpreted. The concentrations decreased with the updated inventory because the condensation (and coagulation) sink has become higher during the previous time steps. Thus, nucleating precursors could have depleted beforehand so that the total NPF rate has become to the level of only one third of the rate simulated with the original inventory and thus giving lower concentrations for the time range in question. Additionally, the neighboring grid cells have even higher sinks; thus, the effect can also be transported from nearby areas. This discussion is now included in the manuscript with the following text added after the discussion of Fig. 7:

"However, the sinks are actually $\sim 4\%$ lower with the updated inventory during the time range presented in Fig. 7. Instead, the sinks just before the time range were $\sim 6\%$ higher and even $\sim 10\%$ higher in an adjacent grid cell. The effect of increased sinks with the updated inventory on the appearance of small particles is not always observed within a single time step or grid cell but within later time steps or nearby grid cells instead, due to a history effect and transportation of components between the grid cells."

[Figure]

**Figure FR6.** Ratios of monthly means of (**a**) the condensation sink (CS) and (**b**) the coagulation sink (CoagS) simulated with the updated and with the original emission inventory.

**20.** *Additionally, related to Fig 7c, I wonder why the modelled PSD jumps so much up and down between different size ranges: the difference in concentrations in neighbouring size bins can be up to two orders of magnitude and the nucleation, Aitken and accumulation mode do not show any modal distribution. Also in Fig 6., the modelled size distributions are surprisingly unsmooth for monthly means.*

The model has internally a tendency to output PSDs in which concentrations in adjacent size bins are clearly different. For better readability, Figs. 6, 7, and 8 are now reproduced using smoothing from the adjacent bins. Nucleation, Aitken, and accumulation modes are also seen more clearly now, especially in Fig. 7c.

**21.** *Lines 355-356: Is the line for observations in London missing, or are there no observations? Why wouldn't the authors use Kumpula observations instead or additionally to London as an example of site with traffic in vicinity? London is not even mentioned in Section 2.2. If there is no data from London, adding Kumpula (or Melpitz) data becomes even more crucial.*

Although there are no suitable observational data for London, it was selected because it was a good example of high traffic densities and low NPF rates (during mornings). It's a good suggestion to use a location for which observational data are available instead of London. Therefore, the data for London in Fig. 8a is now replaced with the data corresponding to Melpitz (the observed distribution is also shown), where similar effects to London are seen too. Melpitz was selected due to high traffic densities nearby. Kumpula would not be as good example because it is located in a grid cell including also sea areas and because photochemical activity—and thus NPF—begins earlier in May due to its northerner location.

**22.** *Language overall: There are many quite long and difficult sentences, due to which I suggest the authors to doublecheck the language in general. Additionally, the use of the word "unity" instead of "one" or "one-to-one" in "ratio over or below unity" does not sound good to me. In my understanding unity is something that cannot be exceeded.*

The language has now been doublechecked. Several long sentences are now revised and split into multiple sentences. The words "unity" are now replaced using the words "one".

**Referee comment 2 (Hugo Denier van der Gon):**

**23.** *The study describes an elegant empirical way to update the only European size-resolved Particle Number inventory and evaluates the results against observations .Ultra fine particles (UFP) and particle numbers (PN) in the atmosphere are not regulated like PM2.5 or PM10. There are no air quality limit values nor obligations to monitor these metrics. As a result our understanding, emission data, measurements, concentrations and related information is scarce. Clearly more information is needed and the paper is a welcome and original contribution, and fitting for ACP. In my opinion the paper can be published after some corrections and further clarifications have been made.*

*Major points*

**24.** *In the introduction first & second paragraph it would be good to already stress/explain that the EUCAARI inventory by definition considers only particles > 10 nm. The reason being that for many emission sources size-resolved PN measurements are extremely scarce, and EFs for <10 nm are either non-existent or highly variable. Literature is also often not clear about the cut-off. To have a more robust result the EUCAARI inventory was made for size bins of 10 nm and up. However, as pointed pout by the authors, this does not mean that the PN , 10 nm are not important and it is very relevant to investigate this. Therefore "updating" mostly means extending the range of the inventory to include an important but difficult size range.*

The second paragraph is now updated to include the sentence:
   "Only emissions of particles larger than 10 nm were estimated in the EUCAARI inventory, because emissions of especially sub-10 nm particles for many emission sources have not been determined with high enough certainty or not determined at all."

**25.** *It is also suggested to explain a bit better that the road transport regulation for PN > 23 nm for non-volatile PN was chosen to have a reproducible measurement. For a standard this is a great advantage. However, a large part of the emitted PN are volatile and it is questionable if this standard for non-volatile PN > 23 nm has any relation or even correlation with the real world total (volatile + non-volatile) size -resolved PN emissions. This relates also to L 301 where some remarks are made about "unregulated vehicle -emitted particles" but volatile particles > 23 nm are also unregulated. So, that is not well-defined in the MS because it suggests road transport particles > 23 nm are regulated but it is only a fraction of these.*

This is now explained better in the introduction by replacing the text:
   "...the fact that only non-volatile particles larger than 23 nm are currently regulated in number emission standards (Giechaskiel et al., 2012) and that the emission factors (EFs) of the smallest particles are quite variable across the vehicle fleet. A high level of variation is caused by the nature of the nucleation process—the main origin of the smallest particles at least in diesel exhaust—which is very sensitive to several factors..."
   with the text:
   "...the fact that only non-volatile particles larger than 23 nm have been selected as the regulated ones in current road transport number emission standards (Giechaskiel et al., 2012) because measuring them is far more reproducible than of volatile ones. Many of the components of the smallest particles do, however, evaporate when heated. Hence, there are also emissions of particles larger than 23 nm (volatile ones) which are currently unregulated. The emission factors (EFs) of the smallest particles are quite variable across the vehicle fleet due to the nature of the nucleation process—their main origin at least in diesel exhaust—which is very sensitive to several factors..."
   The text at the line 301 is also updated to "$N_{<23}$ (**totally** unregulated vehicle-emitted particles, $D_{\mathrm{p}} < 23\,\mathrm{nm}$)" to clarify that the sub-23 nm particles are totally unregulated (both volatile and non-volatile parts).

**26.** *Somewhere in the introduction explain more clearly why you choose to do simulations for a year (2008) that is by now 12-13 years in the past. It is not trivial. Many things have changed by now especially in road transport but also e.g. shipping with fuel sulphur regulations.*

Simulations were done for May 2008 because this period has become a kind of a standard period for PMCAMx-UF simulations for the European domain and is thus used in many other studies as well. Therefore, the model inputs for emissions and meteorology were directly available for that period already. The end part of the introduction is now updated to:

"The simulated period **(May 2008)** was photochemically relatively active, which elevates NPF to the major source of new particles. **This period was chosen because the same period has been simulated in several other related studies as well, providing plenty of comparable data and pre-defined input files for emissions and meteorology. Since the street canyon measurements were performed in 2017—using more recent technologies for PSD measurements—trends of urban aerosol and vehicle emissions were used to scale the determined emissions from 2017 to 2008.**"

**27.** *L186 The updating process was stopped at 57 nm, meaning no changes for Dp > 57 nm. Many studies investigate ultrafine particles which are defined as < 100 nm. Although not the subject of study it would be good to state in discussion or conclusion what the impact of the update is for anthropogenic UFP emissions, especially for road transport. How much does UFP increase? This may help to put in perspective with other studies.*

The impacts of the update for the road transport-related emissions and for the total emissions, in respect to UFP number, are now calculated and included in the text in Sec. 3.2.4 with the following sentence:

"By considering only the number concentrations of ultrafine particles (UFP, sub-100 nm particles), the road transport-related emissions were increased to a 28-fold level. This resulted in that the total UFP number emissions were increased by a factor of 3.1."

Table S1, representing the ratios of monthly means of concentrations simulated with the updated and with the original inventory, is now updated to include also the ratios for the number concentration of UFPs ($N_{<100}$). Whereas the mean increase was $0.9\%$ for the total particle number, it was $1.1\%$ for the UFP number.

**28.** *L190 – why was the scaling done using measurements from 2015 onwards. It would also be possible to use the trend in reported PM2.5 emissions from Finland for road transport exhaust (available at https://www.ceip.at/). Does that give different results? Or do you have a motivation why that would not work?*

Although not covering the years between 2008 and 2015, the trend for PM2.5 was applied because the primary objective was to scale the soot modes onto the same levels. In this way, we were able to consider only updating of the shape of the emitted PSD and to omit the updating of the level of emissions overall. This is now clarified in the manuscript; additionally, an estimation of the realism of the trend of $-7.1\%\mathrm{a}^{-1}$, using the trend of the concentration of 56 nm-sized particles measured in Kumpula between years 2008 and 2017 and the trend of road transport-emitted PM2.5 in Finland reported by EMEP (2021) for years between 2008 and 2017, is now added.

The following text is now added to Sec. 3.2.2:

"The yearly decrease rate of PM2.5 ($7.1\%\mathrm{a}^{-1}$) was, however, reported as statistically not a significant trend (Luoma et al., 2021) and also it only covers the trend between years 2015 and 2018. Thus, a trend was also estimated with the data from Kumpula, which fully cover the years between 2008 and 2017. Applying a seasonal Mann-Kendall test and Sen's slope estimator—as done by Luoma et al. (2021)—to the particle number concentration at 56 nm gives the yearly decrease rate of $4.4\%\mathrm{a}^{-1}$ for the years between 2008 and 2017. Since this trend is for Kumpula, the trend for Mäkelänkatu could be around $7.1\%\mathrm{a}^{-1}$ because the trends of other quantities for Mäkelänkatu were found to be approximately 2-fold than for Kumpula in the study by Luoma et al. (2021). Additionally, the PM2.5 trend was calculated from the data of yearly (1990–2019) road transport emissions (without road, tyre, and brake wear) in Finland, reported by EMEP (2021). The decreasing trend calculated for the years between 2008 and 2017 is $6.0\%\mathrm{a}^{-1}$, which corresponds relatively well to the trend applied here ($7.1\%\mathrm{a}^{-1}$)."

and the following text to Sec. 3.2.3:

"Nevertheless, the scaling of the soot modes was a primary objective here because, hence, the update of the inventory considers only updating the shape of the emitted PSD (below 57 nm), but not its level overall."

**29.** *In the conclusions e.g. around L 415 it would be important to stress / mention that road transport is not the only anthropogenic source where such an update (adding the smaller particles ) would havean impact. I would expect that for aviation (airports close to a city) and shipping (in the case of port cities) this would further increase the PN emissions. You do not have to add these emission here but it is good to mention that this should be addressed as well. How would further addition influence your model results? Is there room for this or would it lead to overestimating?*

It is possible—and even likely—that other anthropogenic sources, such as aviation and shipping activities, may involve underestimations of small particles similar to road transport due to similar reasons. There also seems to be room for adding particle emissions for those activities because, e.g., measured PSDs are higher than the simulated ones for Kumpula, which is located near ($< 15\,\text{km}$) a busy airport and harbor areas. The following text is now added to Sec. 3.3.3:

"Additionally, there are a busy airport and harbor areas within a radius of 15 km from the Kumpula station. It is certainly possible that, in addition to road transport, other activities, such as aviation and shipping, can also involve underestimated particle emissions. Hence, other anthropogenic particle emission sources may also need to be addressed better in emission inventories, in order to have the simulated PSDs to agree with the measured ones."

and the following sentence to the last paragraph of the manuscript:

"In addition to road transport, other anthropogenic emission sources, such as aviation and shipping activities, may need to be addressed better in emission inventories, because they may involve underestimated particle emissions as well."

**30.** *L 345 and further: Do you think having substantial sub-10 nm BC particles is realistic? As BC is a product of incomplete combustion it seems not very likely to me? You state it could also be other non-volatile components such as metals but if that is the case, isn't it better to call it non-volatile instead of BC?*

We agree that soot particles are usually larger than 10 nm. Because the composition of sub-10 nm particles are quite unknown and because we did not want to add an extra component to the inventory (would have required several modifications to the model code), we decided to lump the non-volatile part together with BC. This is now better explained in the text and BC in the updated inventory is now called BC*. Calling it "non-volatile" would not be the best option because there are also other components in the inventory which are non-volatile. The following sentence in Sec. 3.2.4:

"The non-volatile part is here assumed to be BC due to the lack of more specific information."

is now replaced with the sentences:

"The non-volatile part is here lumped together with BC due to the lack of more specific information on its composition and because adding an extra component would have required several modifications to the model code. BC together with the unknown non-volatile part is abbreviated here to BC*."

and the "BC*" abbreviations are now added into legends of Figs. 2b and 7a,d. Also the discussion related to Fig. 7 in Sec. 3.3.4 is now updated to include the "BC*" abbreviations.

**31.** *Moreover & related, I find this "based on results from one diesel bus" (L82) rather tricky in the light of the whole study. How representative is one Finnish diesel bus for the whole European fleet? Why do the authors feel that is good enough? I think this needs better discussion and motivation.*

Please refer to comment 5 (from the another referee) and our reply to it (begins on page 3 of this document) which is a similar comment to this one.

**Minor points / corrections**

**32.** *L38 regulated in* **road transport** *emission standards [it is better to be specific here]*

The text:
   "only non-volatile particles larger than 23 nm are currently regulated in number emission standards"
   is now changed to (partially due to comment 25):
   "only non-volatile particles larger than 23 nm have been selected as the regulated ones in current road transport number emission standards"

**33.** *L80 composition of NCA - easier for the reader to have "composition of 1-3 nm sized particles"*

We decided to keep the form "the composition of NCA (volatile and non-volatile fractions)" there because the form "composition of 1-3 nm sized particles (volatile and non-volatile fractions)" could be more easily mixed with chemical composition, which is not the case here. The word "composition" here refers to the fractions of volatile and non-volatile parts of NCA.

**34.** *L82 chemical composition "obtained from a computational fluid dynamics (CFD) simulation" I find that hard to understand. How can you obtain chemical composition from a CFD simulation? Can you rewrite / explain a bit better?*

It is actually an aerosol dynamics model which gives the chemical composition for particles. This model was coupled with a CFD model. This is now explained better with the updated text: "obtained from **a simulation with an aerosol dynamics model coupled with** a computational fluid dynamics (CFD)  **model**"

**35.** *L143 for*  *both*

The word "the" is now removed from the sentence.

**36.** *L170 and further: The CO2 trick is transparent and elegant but some more discussion or caution on how reliable it is to scale that way to 2008.*

It is true that scaling the emission factors, determined from the data from 2017, to year 2008 needs caution because $CO_2$ emissions from road transport have decreased during the years. Determining the EFs using the $CO_2$ concentrations gives the EFs with respect to kilograms of fuel combusted. Thus, they are applicable to any year. However, the model requires the emission input as time-based particle emission rates. The total amount of fuel combusted has been higher in 2008 than in 2017. Therefore, the emission input would need scaling upwards from year 2017 to year 2008. However, this scaling has already been performed when the EFs were scaled using the trends of PM2.5, because ambient PM2.5 concentrations have decreased not only due to equipping vehicles with a DPF but also due to the fact that the total amount of fuel combusted has decreased. Hence, the scaling of different levels of $CO_2$ emissions has been performed internally already. This discussion is now included in the text with the following paragraph in Sec. 3.2.2:

   "Because fuel efficiency has developed during the years, $CO_2$ emissions from road transport have been on different levels in 2008 and in 2017. The method of determining EFs using $CO_2$ concentrations gives the EFPSD with respect to kilograms of fuel combusted. Therefore, it can be applied to any year. However, the total amount of combusted fuel in the computational grid with respect to time has changed, leading to the need of scaling the time-based particle emission rates—which is the form of the emission input of the model—upwards from year 2017 to year 2008. This scaling has, however, already been performed when the EFs were scaled using the trends of PM2.5, because ambient PM2.5 concentrations have decreased not only due to equipping vehicles with a DPF but also due to the fact that the total amount of fuel combusted has decreased."

**37.** *L174 " the EF of PM2.5 has probably been higher in 2008." Not probably but certainly – you can check the EEA/EMEP emission inventory guidebook for EFs for different EURO classes.*

The text is now updated to: "the EF of PM2.5 has  been higher in 2008 **(EMEP, 2021)**.".

**38.** *L 178 is **the** same*

The word "the" is now added into this sentence.

**39.** *L202 – I am not sure how reassuring the "European average" is . This will be mixing e.g. the UK, Sweden, Bulgaria, Portugal etc. The average may not be very representative of what is seen in the different countries if fleet ages and dominant fuel types are highly variable. On average they may cancel out but PN exposure is about the local urban emissions not about the average emission.*

This is now discussed in Sec. 3.2.3 with the following added text:
   "It should, however, be noted that averaging of vehicle ages or fuel types over Europe is not the most representative in terms of the average emissions or particle exposure because there are countries having old vehicle fleet with mostly diesel vehicles—a combination with a plenty of soot emissions—but also countries having new vehicle fleet also with mostly diesel vehicles—a combination with the least particle emissions. In addition, there are countries with other possible mixtures of fleet ages and fuel types of vehicles as well."

**40.** *Table 1 – Table top row - Please add the size range behind Nucleation and Soot. Easier for the reader.*

We decided to keep the table in its current form because the table already includes count median diameters (CMD) of the nucleation and soot mode (the size range from $D_1$ to $D_2$ for the power law mode). Because the nucleation and soot modes are log-normal distributions, there are, by definition, no lower and upper limits of their particle sizes (except 0.8 nm and 10 μm of the PMCAMx-UF model itself).

**41.** *L229 from the all*

The word "the" is now removed from this sentence as it was probably actually meant with "the".

**42.** *Table 2 – caption "(the more intended one being bold)." This is cryptic – please rewrite e.g. you mean the best performing in bold?*

Yes, "the best performing" was supposed to be meant. It now reads "(the best performing in bold)".

**43.** *L 304 so that  half*

The word "the" is now removed from the sentence.

**44.** *L335 the lie on the range = lie in the range*

The word "on" is now replaced with the word "in" in this sentence.

**45.** *L336 – please rewrite – "travel to human body" is wrong /strange*

The text "can travel to human body" is now replaced with "can thus end up in human body".

**46.** *L 366 after  updating*

The word "the" is now removed from the sentence.

**47.** *L384 "previously underestimated emissions of sub-50 nm particles" I would say "previously partly excluded or partly non-estimated emissions of sub-50 nm particles " The EUCAARI inventory had on purpose a cut-off at 10 nm. In that way it was not "underestimated" but simply not estimated.*

The text is now updated to: "previously  **partly excluded** emissions of sub-50 nm particles".

**48.** *L386 "The reason for the overestimations may be related to overestimated new particle formation" Don't you think that must be related is better? Because the model runs with the original inventory do not include any anthropogenic particles < 10 nm but still give the overestimation?.*

The overestimation may be related to overestimated NPF, but not exactly must be because there is still a possibility that NPF is estimated correctly but the PSD measurements give too low concentrations. Nevertheless, it is sure that the underestimations of sub-10 nm particles are not caused by overestimating their emissions because the overestimations were observed also using the original inventory, in which all sub-10 nm particle emissions were excluded. Additionally, there is a possibility that NPF is estimated correctly but the particle growth out of sub-10 nm size range is underestimated, which was discussed in Sec. 3.3.1 but not in this concluding section. This discussion is now included in the concluding section with the following updated text:

   "The reason for the overestimations may be related to overestimated new particle formation (NPF) **or underestimated particle growth** but also to possibly underestimated particle concentrations from the PSD measurements, which are known to become inaccurate for particle sizes below ∼10 nm. **At least, the overestimations of sub-10 nm particles using the updated inventory are not caused by overestimating their emissions because the overestimations were observed also using the original inventory, in which all sub-10 nm particle emissions were excluded.**"

**49.** *L391 – cryptic – please rewrite*

There was a sentence missing from this text, causing a confusing text. The missing sentence is now added and the following text:
   "There are locations and times having the ratios over and below unity while the mean and median values were slightly over unity. This denotes that the predicted concentrations were increased or decreased with a factor of up to several thousands, depending on the examined particle size range, in certain locations and at certain times after updating the inventory."
   is also updated to finally read:
   "Ratios of simulated particle concentrations after and before updating the inventory were examined from daily and monthly means of local concentrations. The ratios over and below one were observed while the mean and median values were slightly over one: the predicted concentrations were increased or decreased with a factor of up to several thousands, depending on the examined particle size range, in certain locations and at certain times after updating the inventory."

**50.** *L396 total **anthropogenic** particle number (or is that not the case?)*

The word "anthropogenic" is now added to this sentence.

**51.** *L405 replace fuel-combusting vehicles with combustion processes – I don't think the fact that it is a vehicle is important.*

The words "fuel-combusting vehicles" are now replaced with the words "combustion processes" in this sentence.

**52.** *L413 whenever = provided*

[revised manuscript text omitted]

**1 Trimodal fit equation for emission factor particle size distribution**

Emission factor particle size distribution (EFPSD) was fitted to a trimodal distribution with the equation

$$\frac{\mathrm{d}n}{\mathrm{d}\log D_\mathrm{p}} = \left.\frac{\mathrm{d}n}{\mathrm{d}\log D_\mathrm{p}}\right|_\mathrm{power\,law} + \left.\frac{\mathrm{d}n}{\mathrm{d}\log D_\mathrm{p}}\right|_\mathrm{nucleation} + \left.\frac{\mathrm{d}n}{\mathrm{d}\log D_\mathrm{p}}\right|_\mathrm{soot} \tag{S1}$$

where $n$ denotes the emission factor of the particle number belonging to a mode. The first term is expressed with the equation (Olin et al., 2016)

$$\left.\frac{\mathrm{d}n}{\mathrm{d}\log D_\mathrm{p}}\right|_\mathrm{power\,law} = \begin{cases} n\left(\frac{D_\mathrm{p}}{D_2}\right)^\alpha \beta, & D_1 \le D_\mathrm{p} \le D_2 \\ 0, & \text{otherwise} \end{cases} \tag{S2}$$

where $\beta$ is a scaling function

$$\beta\left(\alpha, \frac{D_1}{D_2}\right) = \begin{cases} \dfrac{\alpha\ln 10}{1 - \left(\frac{D_1}{D_2}\right)^\alpha}, & \alpha \ne 0 \\ \dfrac{-\ln 10}{\ln\left(\frac{D_1}{D_2}\right)}, & \alpha = 0 \end{cases} \tag{S3}$$

and $\alpha$, $D_1$, and $D_2$ are the slope parameter and the diameters of the smallest and largest particles of the mode, respectively. The latter two terms are expressed with the equation

$$\left.\frac{\mathrm{d}n}{\mathrm{d}\log D_\mathrm{p}}\right|_\mathrm{nucleation/soot} = \frac{n\ln 10}{\sqrt{2\pi}\ln\mathrm{GSD}} \exp\left[-\frac{\ln^2\left(D_\mathrm{p}/\mathrm{CMD}\right)}{2\ln^2\mathrm{GSD}}\right] \tag{S4}$$

where CMD and GSD are the count median diameter and geometric standard deviation of the mode, respectively. The numerical values of the parameters from the fitting are shown in Table 1.

**2   Description of the CFD-simulations used to determine chemical composition of the emitted particles**

Chemical composition of particles emitted by road traffic was determined using the results from the CFD-simulations by Olin (2013). They consist of a situation where a Euro III bus is driving at a speed of 40 km/h with the engine power of 40 % of the maximum. The situation has been extracted from chasing experiments performed by Rönkkö et al. (2006).

As a Euro III-vehicle, the bus did not have a DPF and it used a diesel fuel having the sulfur content of 50 ppm, which has been the upper limit of the automotive fuel in the EU between years 2004 and 2009.

The simulations were performed using a commercial Ansys Fluent CFD-software with user-defined functions for aerosol dynamics modelling. The CFD-software simulates the flow field (momentum, velocities in 3 dimensions, temperature, and transport of gaseous species) of the bus driving situation including the exhaust flow from the tailpipe. The aerosol model simulates nucleation, condensation, coagulation, deposition, and diffusion of particles emitted directly from the tailpipe and of particles formed from the emitted gaseous precursors after releasing from the tailpipe. The CFD-software also handles the transport of aerosol and connects the aerosol model with temperature and with the concentrations of gaseous species.

Particles were assumed to consist of sulfuric acid ($H_2SO_4$), water ($H_2O$), tetracosane ($C_{24}H_{50}$), and soot. Nucleation was modelled as binary $H_2SO_4$-$H_2O$ nucleation and the nucleation rate was obtained from classical nucleation theory with a correction factor. Condensation modelling included condensation of emitted gases ($H_2SO_4$, $H_2O$, and $C_{24}H_{50}$) on emitted soot particles or on particles formed via nucleation. The concentrations of $H_2SO_4$ and $H_2O$ in the raw exhaust were estimated from fuel and oil sulfur contents and engine parameters. Soot mode properties were obtained from the data measured 10 m behind the bus by chasing. The correction factor for nucleation rate was obtained inversely using measured nucleation mode concentration, i.e., by matching the simulated concentration with the measured one. Particle sizes, instead, were matched with the measured ones using a correction factor for the $C_{24}H_{50}$ concentration, representing the fraction of hydrocarbons which are able to condense on the particles in question.

According to the simulations, particles reached their final sizes (nucleation mode CMD∼10 nm, soot mode CMD∼60 nm) about 5 m behind the bus, but their compositions much earlier. The nucleation mode composition 10 m behind the bus was 11.4 % for $H_2SO_4$, 25.1 % for $H_2O$, and 63.5 % for $C_{24}H_{50}$. The soot mode composition was 6.2 % for $H_2SO_4$, 2.9 % for $H_2O$, 24.1 % for $C_{24}H_{50}$, and 66.8 % for soot. These (without $H_2O$) were used as the mass fractions of sulfate ($SO_4$), primary organic aerosol (POA), and black carbon (BC).

[Figure]

**Figure S1.** Examples of determining emission factors bin-by-bin for three different particle size bins, similarly to the method by Olin et al. (2020). Size-binned particle number concentration data ($dN/d\log D_p$) are averaged within $CO_2$ concentration ($[CO_2]$) bins (circle diameters represent the amount of data used in the averaging). Linear fitting (using the circle diameters as weighting factor) is performed over the averaged data (separately for all 28 measured size bins). The slopes of the linear fits converted to kilograms of fuel combusted are marked in the figure.

[Figure]

**Figure S2.** Monthly means of the particle  mass emission rates of all PMF factors (**a**–**p**).

[Figure]

**Figure S3.** Monthly mean diurnal variations of the particle  mass emission rates in Kumpula/Mäkelänkatu, Finland, and Melpitz, Germany, of all PMF factors (**a–p**).

[Figure]

**Figure S4.** Particle size distributions obtained from PMF factors 6, 7, and 11.

[Figure]

**Figure S5.** Monthly means of the particle mass emission rates (**a**,**c**) from the road transport-related source in the original EUCAARI inventory and (**b**,**d**) from PMF factor 6 (**a**,**b**) as maps and (**c**,**d**) as diurnal variations in Kumpula/Mäkelänkatu, Finland, and Melpitz, Germany.

[Figure]

**Figure S6.** Simulated versus observed number concentrations of particles (**a, c**) smaller than 10 nm ($N_{<10}$) and (**b, d**) larger than 10 nm ($N_{>10}$) at the stations with the highest traffic influences (Melpitz and Kumpula) with (**a, b**) the original and (**c, d**) updated emission inventory. All data correspond to hourly means for May 2008. The solid diagonal lines represent 1:1 lines and the dashed ones 1:2 and 2:1 lines.

**Table S1.** Statistics for grid cell-separated ratios of monthly means of concentrations on the surface-level simulated with the updated and with the original emission inventory. The bold values highlight the particle size ranges experiencing the greatest effects due to updating the inventory.

| | $\dfrac{N_{\mathrm{NCA}}^{\mathrm{upd}}}{N_{\mathrm{NCA}}^{\mathrm{orig}}}$ | $\dfrac{N_{<10}^{\mathrm{upd}}}{N_{<10}^{\mathrm{orig}}}$ | $\dfrac{N_{7-20}^{\mathrm{upd}}}{N_{7-20}^{\mathrm{orig}}}$ | $\dfrac{N_{<23}^{\mathrm{upd}}}{N_{<23}^{\mathrm{orig}}}$ | $\dfrac{N_{<100}^{\mathrm{upd}}}{N_{<100}^{\mathrm{orig}}}$ | $\dfrac{N_{\mathrm{tot}}^{\mathrm{upd}}}{N_{\mathrm{tot}}^{\mathrm{orig}}}$ |
|---|---|---|---|---|---|---|
| Mean | **1.113** | 1.060 | 1.019 | 1.029 | 1.011 | 1.009 |
| Population-weighted mean | 1.100 | 1.053 | **1.096** | 1.034 | 1.023 | 1.022 |
| Median | 0.9996 | 1.001 | **1.007** | 1.0004 | 1.001 | 1.002 |
| Population-weighted median | 1.004 | 1.004 | **1.043** | 1.006 | 1.006 | 1.006 |